# Reconciling In-Context and In-Weight Learning via Dual Representation Space Encoding

**Guanyu Chen**                                    *chen-gy23@mails.tsinghua.edu.cn*
*Department of Automation, Tsinghua University*

**Ruichen Wang**                                   *wang-rc24@mails.tsinghua.edu.cn*
*Department of Automation, Tsinghua University*

**Tianren Zhang**[†]                               *trzhang@mail.tsinghua.edu.cn*
*Department of Automation, Tsinghua University*

**Feng Chen**[†]                                   *chenfeng@mail.tsinghua.edu.cn*
*Department of Automation, Tsinghua University*

**Reviewed on OpenReview:** *https://openreview.net/forum?id=bJK7VIOWAU*

## Abstract

In-context learning (ICL) is a valuable capability exhibited by Transformers pretrained on diverse sequence tasks. However, previous studies have observed that ICL often conflicts with the model's inherent in-weight learning (IWL) ability. By examining the representation space learned by a toy model in synthetic experiments, we identify the shared encoding space for context and samples in Transformers as a potential source of this conflict. To address this, we modify the model architecture to separately encode the context and samples into two distinct spaces: a *task representation space* and a *sample representation space*. We model these two spaces under a simple yet principled framework, assuming a linear representational structure and treating them as a pair of dual spaces. Both theoretical analysis and empirical results demonstrate the effectiveness of our proposed architecture, CoQE, in the single-value answer setting. It not only enhances ICL performance through improved representation learning, but also successfully reconciles ICL and IWL capabilities across synthetic few-shot classification and a newly designed pseudo-arithmetic task. The code is available at: `https://github.com/McGuinnessChen/dual-representation-space-encoding`.

## 1 Introduction

In recent years, large-scale models based on the Transformer architecture have demonstrated remarkable capabilities across language (Brown et al., 2020; Guo et al., 2025), vision (Achiam et al., 2023; Maaz et al., 2024), and robotics (Driess et al., 2023; Zitkovich et al., 2023). Among these capabilities, the in-context learning (ICL) ability has drawn increasing attention, as it offers a general paradigm for task generalization. ICL refers to the capability of a pretrained Transformer model to solve previously unseen tasks by using demonstration examples in the prompt—without updating its parameters. In contrast, in-weight learning (IWL) characterizes the conventional ability of a model to recall the memory stored in weights. An ideal model would seamlessly integrate both capabilities: relying on memory to handle training tasks, while adapting to new tasks through contextual cues.

However, recent studies suggest that there exists an inherent conflict between ICL and IWL (Park et al., 2025; Nguyen & Reddy, 2025). This leads to a notable performance degradation when the demonstration examples deviate from the training distribution (Chan et al., 2025), thereby limiting the generalization ability

---

[†]Co-corresponding authors.

of ICL. This conflict is closely related to training settings such as data distribution (Chan et al., 2022), model size (Shi et al., 2024), and training time (Singh et al., 2025). Singh et al. (2023) hypothesize that it arises from the competition between the two capabilities for the shared model circuits. Nguyen & Reddy (2025) on the other hand, attributes this to the different relative learning rates of ICL and IWL. Furthermore, some studies (Chan et al., 2022; Singh et al., 2023; Anand et al., 2025) have attempted to alleviate the tension between ICL and IWL, but can only achieve a tradeoff under particular Zipfian data distributions. Hence, it remains a valuable question to understand and eliminate the conflict between ICL and IWL.

In this work, we aim to understand why models tend to oscillate between ICL and IWL capabilities under different settings from the perspective of representation learning. We observe that in models with strong IWL performance, the learned representation space provides a sufficiently rich representation of the query samples themselves. In contrast, models with strong ICL performance learn representation spaces that effectively encode contextual information. However, these two desirable structures of representation are difficult to acquire simultaneously, making it challenging to achieve strong ICL and IWL performance at the same time (see Figure 1 bottom). Motivated by this observation, we hypothesize that the ICL-IWL conflict can be alleviated by encoding the context and query samples into separate spaces, thereby preventing interference between the two types of information. We refer to these spaces as the *task representation space* and the *sample representation space*, respectively. To provide a principled modeling of these two spaces, we build upon the widely accepted linear representation hypothesis (Mikolov et al., 2013; Nanda et al., 2023; Park et al., 2024) and propose to model the task representation space as the dual space of the sample representation space. We prove the completeness of a sample representation space under sufficient training tasks, which could facilitate task generalization by ICL. We also formalize the entangled nature of Transformers' encoding process – standard softmax attention does not support such a dual-space structure, highlighting a contrast with linear attention mechanisms commonly adopted in recent theoretical analysis.

Furthermore, we propose a straightforward architectural design, CoQE, as an algorithmic exploration. Unlike standard Transformers, CoQE employs separate pathways to encode context and query samples, aiming to learn the task and sample representation space, respectively. The final model output is obtained by computing the inner product between elements from the two spaces according to the Riesz representation theorem. Our synthetic experimental results show that CoQE not only achieves lower ICL error in both in-distribution and out-of-distribution scenarios, but also reconciles ICL and IWL across synthetic few-shot classification task of Chan et al. (2022); Singh et al. (2023; 2025) and our newly designed pseudo-arithmetic task.

## 2 Related Work

**Investigations on ICL Mechanisms.** From the theoretical perspective, existing works have examined how Transformers perform ICL across various scenarios (Zhang et al., 2024a; Li et al., 2023; Tian et al., 2023; Nichani et al., 2024; Chen et al., 2024; Wu et al., 2024; Huang & Ge, 2024; Oko et al., 2024; Liang et al., 2025). These studies typically analyze simplified architectures such as linear self-attention or query-key-combined formulations, and many of them conduct analyses from the perspective of Bayesian model averaging (Zhang et al., 2025b; Ye et al., 2024). From the empirical perspective, Garg et al. (2022) firstly demonstrated that Transformer-based ICL can generalize effectively to out-of-distribution (OOD) tasks, leading to a surge of interest in exploring its generalization behavior (Ahuja & Lopez-Paz, 2023; Kossen et al., 2024; Pan et al., 2023; Fan et al., 2024; Xiong et al., 2025; Yadlowsky et al., 2023; Bu et al., 2025). Some work has shown that LLM can implicitly encode task vectors during ICL (Hendel et al., 2023; Todd et al., 2024; Guo et al., 2024; Yang et al., 2025; Han et al., 2025). Unlike these works that focus only on task information encoded in the model's representation space, we examine the relationship between task information and the intrinsic information of the samples. From this perspective, we seek to explain the conflict between ICL and IWL.

**Relationship between ICL and IWL.** Beyond investigations on the ICL mechanisms, some studies have found that ICL is not a guaranteed and stable capability of Transformers; rather, it competes with the model's inherent in-weight learning (IWL) ability, which relies on information stored in the weights (Chan et al., 2022; Singh et al., 2023; Reddy, 2024; Panwar et al., 2024). Chan et al. (2022) examined the impact of different training data distributions on both abilities, finding that only when the training data follows a certain Zipfian distribution can both abilities coexist. Singh et al. (2023) further confirmed the transient

nature of ICL, observing that it always fades after emerging and gives way to IWL. Nguyen & Reddy (2025) attributes this to the different relative learning rates of ICL and IWL, and conducted an analysis on a simplified one-layer transformer model. Chan et al. (2025) proposed a simple theoretical model, which is a linear combination of an in-weight learner and an in-context learner. Singh et al. (2025) empirically discovered a more complex coopetition relationship between ICL and IWL. However, to date, no work has truly achieved coexistence between ICL and IWL across varied training conditions.

**Linearization in Latent Space.** Beyond task-specific vectors, a line of work has examined how models internally encode a variety of abstract concepts as linear vectors in latent space, giving rise to the commonly accepted *linear representation hypothesis* (Mikolov et al., 2013; Nanda et al., 2023; Park et al., 2024). Several studies have shown that concepts such as truthfulness (Marks & Tegmark, 2024), time and space (Gurnee & Tegmark, 2024), and other semantic properties (Dalvi et al., 2022; Merullo et al., 2024; Ye et al., 2025) can emerge in the model's latent space, using linear probes as the primary tool. Additionally, larger models tend to yield more disentangled and interpretable internal representations (Bricken et al., 2023; Cunningham et al., 2023), and this can be regarded as evidence of a world model within large models (Zhang et al., 2025a). In this work, to provide a principled treatment of the encoding spaces of context and query samples, we introduce the notion of a linear task representation space, grounded in the linear representation hypothesis.

## 3 Preliminaries

**In-context learning setup.** The basic setup for analyzing ICL was first introduced by Garg et al. (2022) and has since been widely adopted (Yadlowsky et al., 2023; Pan et al., 2023). Consider a distribution $\mathcal{D}_{\mathcal{X}}$ over an input space $\mathcal{X} \subseteq \mathbb{R}^{d_x}$, and let $\mathcal{F}$ denote a class of functions over a distribution $\mathcal{D}_{\mathcal{F}}$. For each prompt, we first sample a task $f \sim \mathcal{D}_{\mathcal{F}}$, then draw a set of $n$ input-output pairs $\{(\mathbf{x}_i, y_i)\}_{i=1}^n$, where $\mathbf{x}_i \overset{\text{i.i.d.}}{\sim} \mathcal{D}_{\mathcal{X}}$ and $y_i = f(\mathbf{x}_i)$. These sample pairs serve as context demonstrations. Then, we independently generate a query input $\mathbf{x}_q \sim \mathcal{D}_{\mathcal{X}}$. The final prompt is gathered as a sequence:

$$\mathcal{P} = \big(\mathbf{x}_1, y_1, \ldots, \mathbf{x}_n, y_n, \mathbf{x}_q\big).$$

The ICL capability of a pretrained model $\mathbb{M}_\theta$ refers to its accuracy to produce predictions $\hat{y}_q = \mathbb{M}_\theta(\mathcal{P})$ for $y_q = f(\mathbf{x}_q)$, without having explicit knowledge of the current task $f$ and without updating its parameters.

Chan et al. (2022) extend this setting by introducing few-shot image classification tasks. In this setup, $\mathbf{x}$ represents an encoded image, and $\mathcal{F}$, as a set of classifiers, maps $\mathcal{X}$ to a finite label set $\mathcal{Y}$. The ICL capability refers to the model's ability to correctly classify a query image $\mathbf{x}_q$ based on the demonstrations.

**Transformer model.** A standard single-head self-attention (SA) layer (Vaswani et al., 2017) operates on an input matrix $Z \in \mathbb{R}^{d_e \times L}$, where $L$ is the sequence length and $d_e$ the embedding dimension. Let $Q = W_Q Z$, $K = W_K Z$, $V = W_V Z$ with $W_Q, W_K \in \mathbb{R}^{d_k \times d_e}$ and $W_V \in \mathbb{R}^{d_v \times d_e}$. The attention output is

$$\mathrm{SA}(Z) = Z + W_O\, V \cdot \mathrm{softmax}\left(\frac{K^\top Q}{\sqrt{d_k}}\right),$$

where $W_O \in \mathbb{R}^{d_e \times d_v}$ and the softmax is applied column-wise.

For the theoretical analysis of ICL, the prompt $\mathcal{P}$ is typically re-organized into an embedding matrix:

$$Z = \begin{pmatrix} \mathbf{x}_1 & \cdots & \mathbf{x}_n & \mathbf{x}_q \\ y_1 & \cdots & y_n & 0 \end{pmatrix} \in \mathbb{R}^{(d_x+1)\times(n+1)},$$

where $d_x$ is the input feature dimension. Moreover, they often use a linear self-attention (LSA) variant obtained by removing the softmax and merging parameters:

$$\mathrm{LSA}(Z) = Z + \frac{1}{n} W_{OV} Z Z^\top W_{KQ} Z,$$

where $W_{OV} = W_O W_V$, $W_{KQ} = W_K^\top W_Q \in \mathbb{R}^{(d_x+1)\times(d_x+1)}$ are trainable, and $1/n$ is a scaling constant. The model prediction $\hat{y}_q$ for the query is taken as the bottom-right entry of $\mathrm{LSA}(Z)$.

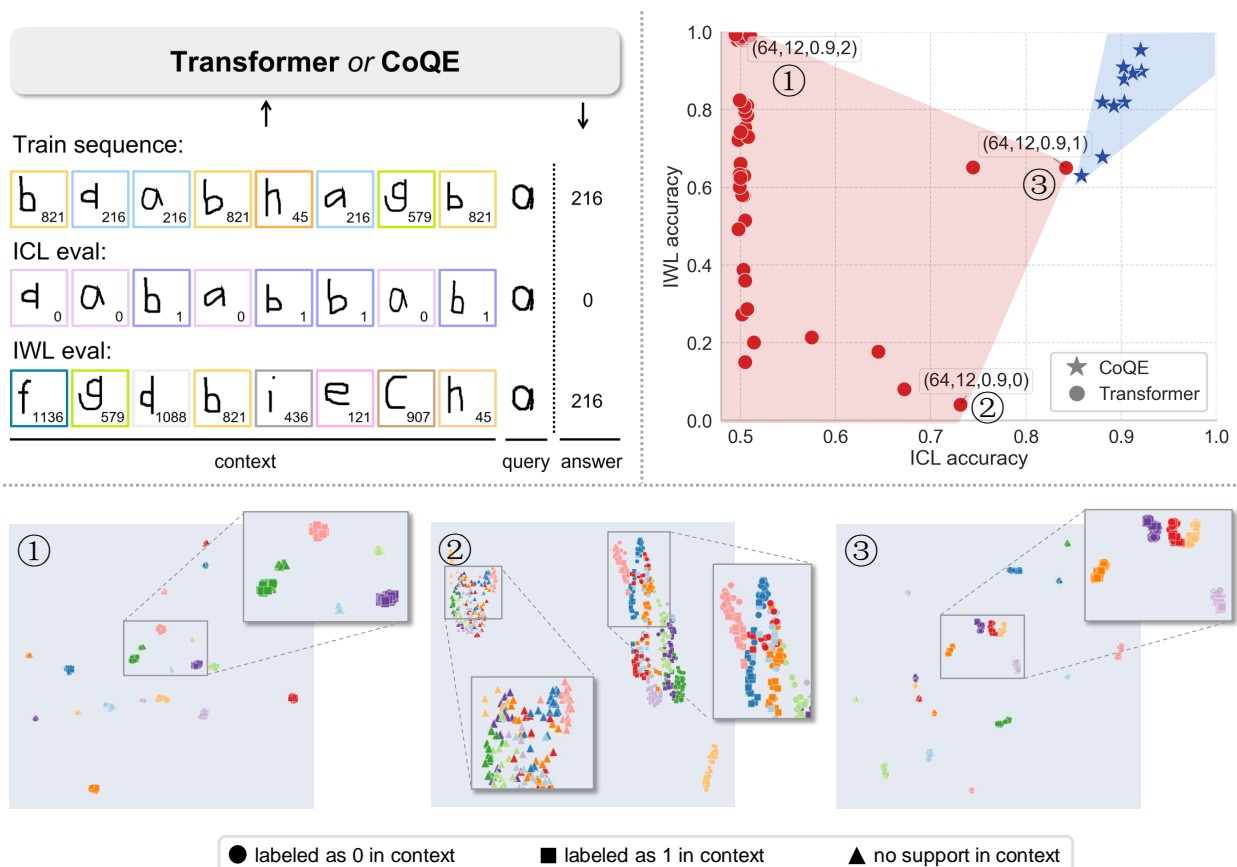

Figure 1: (Top left) Overview of our synthetic task setup. (Top right) ICL and IWL performances under different training settings ($E$, $L$, $P_{\text{bursty}}$, $\alpha$). The Transformers fluctuate between ICL and IWL capabilities, whereas our CoQE models robustly reconcile the two capabilities. (Bottom) Representation visualization of ten classes on distinct context conditions. We observe that good clusters for samples and good clusters for contexts are hard to achieve simultaneously. Detailed discussions are presented in Section 4.

**Dual space.** Before formally introducing our dual-space modeling framework, we first present the general mathematical definition of the dual space.

**Definition 3.1** (Dual space). *Let $V$ be a finite-dimensional inner product space over a field $\mathbb{F}$ (typically $\mathbb{R}$ or $\mathbb{C}$) with inner product $\langle \cdot, \cdot \rangle$. The dual space of $V$, denoted $V^*$, is the set of all linear functionals from $V$ to $\mathbb{F}$:*

$$V^* \triangleq \{f : V \to \mathbb{F} \mid f \text{ is linear}\}. \tag{1}$$

For every $f \in V^*$, there exists a unique vector $\omega \in V$, called the Riesz representation of $f$, such that $f(v) = \langle \omega, v \rangle, \forall v \in V$. Let $\{e_1, \ldots, e_n\}$ be the basis of $V$. The dual basis $\{e^1, \ldots, e^n\} \subset V^*$ is defined by $e^i(e_j) = \delta_{ij}, 1 \leq i, j \leq n$, where $\delta_{ij}$ is the Kronecker delta.

In the following, we will show that this dual-space formulation can be used to model the relationship between a task representation space and the model's sample representation space. Moreover, by the Riesz representation theorem, elements from the two spaces can be composed via inner product.

## 4    Representation Space Analysis

In this section, we train and evaluate Transformers on synthetic datasets (Figure 1 top left) under a range of training conditions. By examining the learned representation spaces, we investigate how models with good ICL performance differ from those that excel in IWL.

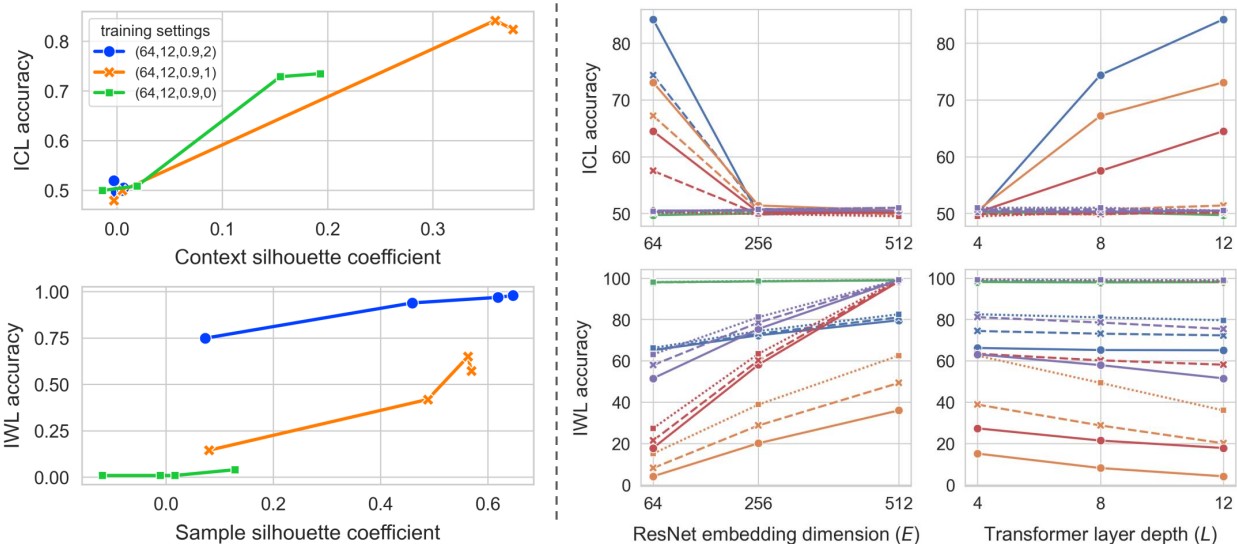

Figure 2: (Left) For three training settings and four training checkpoints (3k, 10k, 50k, 100k steps), there are clear positive correlations between ICL performance and CSC, and between IWL performance and SSC. (Right) Through various experiments, we observe that larger $E$ improves IWL but causes ICL to disappear, and larger $L$ consistently enhances ICL while slightly harming IWL. Different colors and line styles represent different training settings. Detailed results are provided in Table 3.

## 4.1 Synthetic Task Setup

We use a synthetic few-shot classification task based on the Omniglot dataset (Lake et al., 2015). The dataset contains $1,623$ character classes, each with 20 samples. We construct prompt sequences that each consists of eight image-label pairs followed by a query image (Chan et al., 2022; Singh et al., 2023; 2025). We use a ResNet to encode images input, and the training objective minimizes the cross-entropy loss between the model's prediction and the correct label for the query image.

Training sequences have two key properties that affect the tradeoff between ICL and IWL: burstiness and Zipfian exponent. In bursty sequences, three out of the eight image-label pairs in the context share the same class as the query sample. This setup allows the model to infer the correct label based on context alone, which has been found to incentivize ICL while suppressing IWL (Chan et al., 2022). To avoid repetition biases, bursty sequences additionally include three image-label pairs from a distinct distractor class. $P_{\text{bursty}}$ denotes the proportion of bursty sequences in the training set, while the rest are generated via random sampling. The second factor is the Zipfian exponent, which controls the frequency distribution of different classes. Under the Zipfian distribution, the class probability is defined as $p(R = r) \propto 1/r^{\alpha}$, where $R$ is the rank of the class, and $\alpha$ is the Zipfian exponent. When $\alpha = 0$, the distribution becomes uniform. Chan et al. (2022) observe that when $\alpha = 1$, a *sweet spot* emerges, where the model reaches a tradeoff for both ICL and IWL. During training, we keep image-label mappings fixed.

Test sequences are divided into two kinds, corresponding respectively to the evaluation of ICL and IWL capability. For ICL, we use sequences with four images from each of two classes, and we set the class labels to either 0 or 1 randomly for each sequence. Accuracy on this evaluator is measured across 0 and 1 as possible outputs, and chance-level accuracy is 50%. As these labels are not associated with these images during training, the only way to achieve above-chance accuracy is to refer back to the context. For IWL, we use sequences where none of the context images come from the same class as the query, but all of the image-label mappings are the same as during training. In this case, ICL is not useful, as there are no matching images in context, so the model must rely on mappings stored in weights.

## 4.2 Observations and Analysis

During training, we consider four factors: the ResNet embedding dimension $E$, the number of Transformer layers $L$, the bursty proportion $P_{\text{bursty}}$, and the Zipfian exponent $\alpha$. The training outcomes under different

combinations of these factors are shown in the red region of Figure 1 top right. We also collect and visualize the query sample representations of three models on the boundary of the red region, corresponding to the configurations $(64, 12, 0.9, 2)$, $(64, 12, 0.9, 0)$, and $(64, 12, 0.9, 1)$. Specifically, we examine the token embeddings of $\mathbf{x}_q$ under three different context conditions: (1) when four images of the same class as $\mathbf{x}_q$ appear in the context and are labeled as 0, (2) when they are labeled as 1, and (3) when no image of the same class appears in the context. For the choice of token embedding layer, we empirically find that different layers yield consistent qualitative results; therefore, we directly use the final layer embeddings. We focus on the ten most common query classes, and the UMAP visualizations are shown in the Figure 1 bottom.

**Observation 1: ICL corresponds to good representations for context, while IWL corresponds to good representations for samples. The two are difficult to achieve simultaneously.** As shown in Figure 1 bottom, the first model forms well-clustered representations of query samples from different categories (points with different colors), but fails to capture the label distinctions provided in the context demonstrations (the squares and circles of the same color overlap). This indicates that the learned representation space sufficiently encodes intrinsic information of samples but lacks a clear encoding of the task information contained in the demonstrations. Consequently, the model achieves high IWL accuracy but exhibits almost no ICL capability. In contrast, the second model shows the opposite behavior: it fails to form distinct clusters for samples from different categories but demonstrates sensitivity to contextual information (the squares and circles of the same color are shifted apart). So the model has little IWL ability but possesses some ICL capability. The third model achieves a tradeoff between the sample information and contextual information in its representation space, thereby realizing a tradeoff between ICL and IWL performance as well. However, the training conditions are rather restrictive, and both abilities remain suboptimal.

To further support our observation, we employ two quantitative metrics derived from the standard silhouette coefficient, which evaluates cluster cohesion and separation. The silhouette score $s(x)$ measures how similar a sample $x$ is to its own cluster compared to other clusters. Using Euclidean distance $d(\cdot, \cdot)$, we define $a(x)$ as the average distance between $x$ and all other points in its own cluster $C_x$, and $b(x)$ as the average distance to the nearest neighboring cluster $C_k$ (where $k \neq x$).

$$a(x) = \frac{1}{|C_x| - 1} \sum_{x' \in C_x, x' \neq x} d(x, x'), \quad b(x) = \min_{C_k \neq C_x} \left( \frac{1}{|C_k|} \sum_{x' \in C_k} d(x, x') \right) \tag{2}$$

The silhouette score is then given by:

$$s(x) = \frac{b(x) - a(x)}{\max\{a(x),\, b(x)\}} \tag{3}$$

The first metric we employ is context silhouette coefficient (CSC). It measure whether the model learns "good representations for context". For a fixed object class $c \in \{1, \ldots, N\}$ (with $N = 10$), we treat the samples as two distinct clusters based on their contextual labels (0 or 1). Here, $a(z)$ measures the distance to samples with the same context label, while $b(z)$ measures the distance to the opposite context label. We average the scores within each class $c$ and then across all $N$ classes:

$$\text{CSC} = \frac{1}{N} \sum_{c=1}^{N} \frac{1}{|Z_c|} \sum_{z \in Z_c} s(z). \tag{4}$$

The second metric is sample silhouette coefficient (SSC). SSC adapts the metric to measure "good representations for samples" without context support. In this setting, the clusters are defined by the $N$ ground-truth object classes. For a sample $i$ with class $y_i$, $a(i)$ is the intra-class distance, and $b(i)$ is the distance to the closest distinct class. The final metric is the average across all $M$ samples of all classes:

$$\text{SSC} = \frac{1}{M} \sum_{i=1}^{M} s(i). \tag{5}$$

In Figure 2 left, we report how ICL performance correlates with CSC, and how IWL performance correlates with SSC. Both show a clear positive correlation, providing quantitative evidence for our observation that the structure of the learned representation space is tightly linked to the model's ICL and IWL behavior.

**Observation 2: Model size also affects the ICL-IWL tradeoff.** We make another new finding that model size also strongly affects the ICL-IWL tradeoff in standard Transformers (Figure 2 right). We observe that, under the same conditions, a 12-layer Transformer exhibits stronger ICL but weaker IWL compared to a 4-layer Transformer. We suppose that this is due to the Transformer's inductive bias toward attending to context, compared to just memorizing context-irrelevant sample information. We also observe that increasing the ResNet embedding dimension from 64 to 512 nearly eliminates the model's ICL ability while substantially enhancing IWL. Notably, we connect a fully connected layer after the ResNet to reduce the dimension back to 64 before inputting to the Transformer, ensuring that the latter's role remains unchanged. We speculate that the larger ResNet increases the expressivity of individual sample tokens, and when a single token is sufficiently expressive to solve the task, the model tends to ignore the context. This is consistent with Observation 1 that sample information and contextual information are difficult to coexist during training.

## 5 Dual-Space Modeling of Task and Sample Repressentations

We have shown that the conflict between ICL and IWL originates from the difficulty of simultaneously encoding sample information and contextual information within the same representation space of Transformers. We aim to reconcile ICL and IWL by encoding the context and the query samples into separate representation spaces, thereby addressing their incompatibility within a single space. This raises a new question: how can we provide a principled modeling of these two spaces such that they can interact in a coherent and meaningful way? To this end, we start from the simple yet powerful assumption of a linear representational structure and propose our *dual-space* modeling framework. All proofs are provided in the Appendix A.

### 5.1 Task Representation Space

We begin with the widely acknowledged linear representation hypothesis (Mikolov et al., 2013; Nanda et al., 2023; Park et al., 2024), which is the idea that models internally encode abstract semantics of samples as linear vectors in latent space. And because tasks are inherently semantic, this hypothesis can be formalized from a measurement perspective according to Park et al. (2024).

**Definition 5.1** (Linear sample representation space). *Let $\mathcal{X} \subseteq \mathbb{R}^{d_x}$ denote the input space and $\mathcal{Y}$ the label set. A linear sample representation space $\mathcal{M} \subseteq \mathbb{R}^d$ is a finite-dimensional inner product space equipped with a mapping $\phi : \mathcal{X} \to \mathcal{M}$, such that*

1. *(Learnability) $\phi$ is parameterized by a model $\mathbb{M}$ and can be learned from data;*

2. *(Linear Measurement) In the case of regression with $\mathcal{Y} \subseteq \mathbb{R}$, there exists a linear transformation $\omega$ such that, given any $(\mathbf{x}, \mathrm{y})$ pair, the label can be expressed as*

$$\mathrm{y} = \langle \omega, \phi(\mathbf{x}) \rangle. \tag{6}$$

   *In the case of classification with $\mathcal{Y} = \{0, 1\}$, the label probability is given by*

$$\mathrm{logit} \mathbb{P}(\mathrm{y} = 1 \mid \mathbf{x}) = \langle \omega, \phi(\mathbf{x}) \rangle. \tag{7}$$

Definition 5.1 formalizes the notion of a sample representation space under the linear representation hypothesis in the single-task setting. It can be straightforwardly extend to the multi-task case, assuming that *there exists a shared linear sample representation space across different ICL tasks*. Note that this assumption has been implicitly embedded in a wide range of theoretical and algorithmic work (Garg et al., 2022; Hu et al., 2023; Kim & Suzuki, 2024; Zhang et al., 2024b). Based on this assumption, we define the corresponding linear task transformation space. Without loss of generality, we consider only the regression case.

**Definition 5.2** (Linear task transformation space). *Let $\mathcal{F} = \{f : \mathcal{X} \to \mathbb{R}\}$ denote a task function space defined over the input space $\mathcal{X}$. We assume that there exists a sample representation space $\mathcal{M}_{\mathcal{F}} \subseteq \mathbb{R}^d$, together with a mapping $\phi_{\mathcal{F}}$, such that $\mathcal{M}_{\mathcal{F}}$ is linear with respect to $\mathcal{X}$ and each label set $\mathcal{Y}_f = \{f(\mathbf{x}) \mid \mathbf{x} \in \mathcal{X}\}$, $\forall f \in \mathcal{F}$. A linear task transformation space is then defined as a linear functional space $\mathcal{T} = \{t : \mathcal{M}_{\mathcal{F}} \to \mathbb{R}\}$, equipped with a mapping $\psi : \mathcal{F} \to \mathcal{T}$ such that for any $f \in \mathcal{F}$, $\psi(f) = t$ satisfying*

$$f(\mathbf{x}) = t(\phi_{\mathcal{F}}(\mathbf{x})), \quad \forall \mathbf{x} \in \mathcal{X}. \tag{8}$$

Building upon this foundation, our key theoretical contribution is that we find the task transformation space can be modeled as the dual space of the sample representation space. See Appendix A.1 for the proof.

**Proposition 5.3** (Task-sample duality). *Let $\mathcal{X}$ be the input space and $\mathcal{Y}_f$ the multiple label sets corresponding to each task $f \in \mathcal{F}$. Under Definition 5.2, there exists a linear sample representation space $\mathcal{M}_{\mathcal{F}}$ and a linear task transformation space $\mathcal{T}$, where $\mathcal{T}$ is the dual space of $\mathcal{M}_{\mathcal{F}}$, i.e. $\mathcal{T} = \mathcal{M}_{\mathcal{F}}^*$.*

**Definition 5.4** (Task representation space). *Under Proposition 5.3, for each task $f \in \mathcal{F}$, $\psi(f) \in \mathcal{T}$ admits a unique Riesz representation $\omega_f$. The task representation space $\mathcal{W}_{\mathcal{F}}$ is defined as the set of all such Riesz representations. Then for any $f \in \mathcal{F}$, we have*

$$f(\mathbf{x}) = \langle \omega_f, \phi_{\mathcal{F}}(\mathbf{x}) \rangle, \quad \forall \mathbf{x} \in \mathcal{X}. \tag{9}$$

In summary, we map various nonlinear ICL tasks to vectors in the task representation space, leveraging the linear representation hypothesis, the dual-space formulation, and the Riesz representation theorem. From the above formulation, we can further define basis representations and basis transformations.

**Definition 5.5** (Basis task representations). *Under Proposition 5.3, let $\{m_1, \ldots, m_d\}$ be a basis of the sample representation space $\mathcal{M}_{\mathcal{F}}$, and let $\{t_1, \ldots, t_d\}$ be the corresponding dual basis of the task transformation space $\mathcal{T}$. The basis task representations are defined as the Riesz representations of $\{t_1, \ldots, t_d\}$, denoted by $\{\omega_1, \ldots, \omega_d\}$, which satisfy*

$$\langle \omega_i, m_j \rangle = \delta_{ij}, \quad 1 \leq i, j \leq d. \tag{10}$$

Thus, every sample representation $\phi_{\mathcal{F}}(\mathbf{x})$ can decompose uniquely as $\phi_{\mathcal{F}}(\mathbf{x}) = \sum_{i=1}^{d} \alpha_i(\mathbf{x}) m_i$, and every task representation $\omega_f$ can decompose uniquely as $\omega_f = \sum_{j=1}^{d} \beta_j \omega_j$. The output can be given by

$$\langle \omega_f, \phi_{\mathcal{F}}(\mathbf{x}) \rangle = \sum_{i=1}^{d} \alpha_i(\mathbf{x}) \beta_i.$$

Our next Theorem 5.6 shows that, under the dual-space modeling framework, a sufficient set of ICL tasks guarantees a basis-covering sample representation space. See Appendix A.2 for the proof.

**Theorem 5.6** (Completeness of basis representations under task traversal). *Under Proposition 5.3, we assume that a learner with sample representation mapping $\phi_\theta$ is presented with a task traversal curriculum $\mathcal{C}$ such that: $\mathrm{span}\{t \mid t \in \mathcal{C}\} = \mathcal{T}$. Then, if the learner achieves zero empirical error, the learned representation mapping $\phi_\theta$ satisfies: $\mathrm{span}\{\phi_\theta(\mathbf{x}) \mid \mathbf{x} \in \mathcal{X}\} = \mathcal{M}_{\mathcal{F}}$.*

It can be expected that, by promoting a basis-covering sample representation space, our dual-space modeling framework could enhance ICL capability. We empirically validate this in Section 7.1.

## 5.2 ICL Formulation under Dual-Space Modeling

In this section, we formulize ICL under our dual-space modeling framework.

**Definition 5.7** (Context-induced task representation in ICL). *In the ICL setting, the task representation can be specified jointly by two components: (1) context demonstration of labeled examples $\mathbf{z}_{1:n} = (\mathbf{z}_1, \ldots, \mathbf{z}_n)$ with $\mathbf{z}_i = (\mathbf{x}_i, \mathrm{y}_i) \in \mathcal{X} \times \mathcal{Y}$, and (2) a representation mapping $\phi : \mathcal{X} \to \mathbb{R}^d$. That is*

$$\omega_f \triangleq \omega_f(\mathbf{z}_{1:n}, \phi). \tag{11}$$

Definition 5.7 formalizes that in the ICL setting, the task specified by a prompt is determined by its context portion. Thus, the encoding space of context naturally serves as the task representation space. We further show that existing theoretical analyses of ICL based on LSA (Zhang et al., 2024a; Kim & Suzuki, 2024; Wu et al., 2024) are fully compatible with our framework, from which we can derive a closed form of $\omega_f$.

**Proposition 5.8** (Closed form of $\omega_f$ under simplified LSA). *Consider an LSA layer applied after a feature encoder $\phi : \mathcal{X} \to \mathbb{R}^d$ implemented by an MLP. Suppose the LSA projection matrices $W_{KQ}$ and $W_{OA}$ are initialized such that*

$$W_{OV} = \begin{pmatrix} * & * \\ 0_d^\top & 1 \end{pmatrix}, \qquad W_{KQ} = \begin{pmatrix} \Theta & 0_d \\ 0_d^\top & * \end{pmatrix}.$$

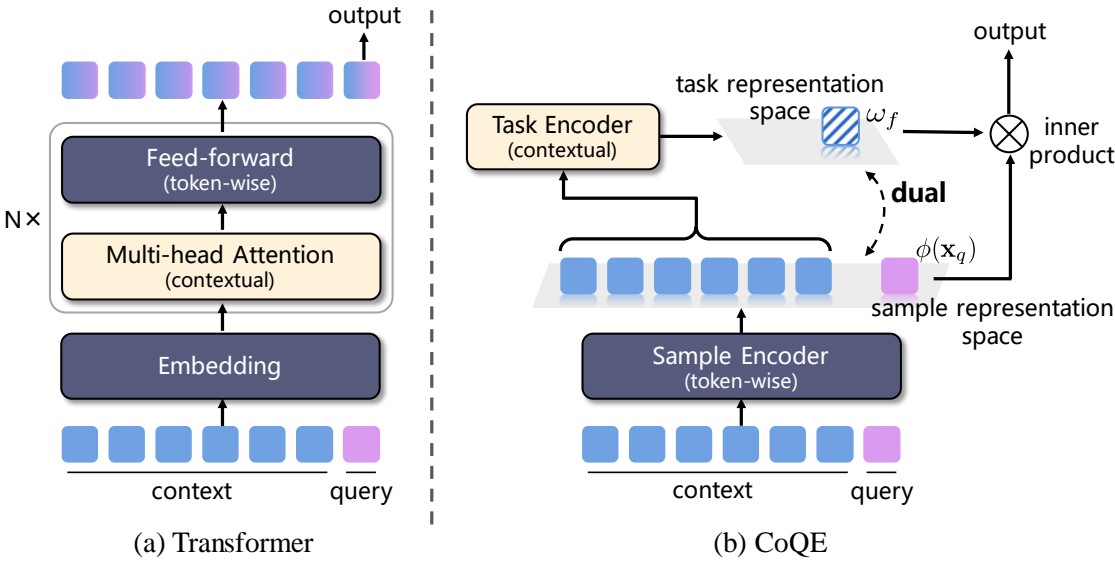

Figure 3: Comparison of Transformer and CoQE architectures. Unlike the Transformer, which encodes both context-level and sample-level information into the same representation space, CoQE implements dual representation spaces encoding that explicitly distinguishes between context and samples.

*Then the final prediction takes the form* $\hat{y} = \langle \omega_f(\mathbf{z}_{1:n}, \phi), \phi(\mathbf{x}_q) \rangle$, *where*

$$\omega_f(\mathbf{z}_{1:n}, \phi) = \frac{1}{n} \sum_{i=1}^{n} \mathbf{y}_i \Theta^\top \phi(\mathbf{x}_i). \tag{12}$$

Proposition 5.8 can explain the effectiveness of LSA simplification in analyzing ICL: it implicitly performs our dual-space modeling between the task representation space and the sample representation space. However, we argue that *it fails to capture the entanglement of standard Transformers encoding progress*, which use the original, unsimplified SA. As we will show in the next theorem, SA cannot realize such dual-space modeling.

**Theorem 5.9** (Entangled structure under general SA)**.** *For a standard SA model with softmax-based attention weights, there does NOT exist a pair of $\phi_0$ and $\omega_0(\mathbf{z}_{1:n}, \phi_0)$, such that the model prediction admits the following decomposition:*

$$\hat{y}_q = \langle \omega_0(\mathbf{z}_{1:n}, \phi_0), \phi_0(\mathbf{x}_q) \rangle. \tag{13}$$

From our dual-space modeling perspective, Theorem 5.9 formalizes the entangled nature of how Transformers encode context and sample-level information. See Appendix A.4 for the proof. We posit that this entanglement is the underlying reason for the observed conflict between ICL and IWL.

## 6 CoQE: A Transformer with Separate Context-Query Encoding

We have established a principled modeling framework for the task representation space and the sample representation space. Next, we propose a straightforward yet effective architectural modification: CoQE, a Transformer with separate **Co**ntext-**Q**uery **E**ncoding.

The CoQE model consists of two modules: a shared sample encoder $\phi_\theta$ and a dedicated task encoder $\omega_\theta$, as shown in Figure 3 (b). The sample encoder generates general-purpose representations for all samples, including the query. We implement it with a token-wise module, for it should process samples independently without considering context. The task encoder, on the other hand, operates on the general representations of the context and focuses on producing the representation of the current task. Thus this module should be contextual and has the capability to condense sequential information. Finally, the prediction output is obtained by computing the inner product between the task representation and the query sample representation. Taking the regression task as an example, the formalization of CoQE output is as follows:

$$\hat{y}_q = \langle \, \omega_\theta \left( \phi_\theta(\mathbf{z}_{1:n}) \right), \, \phi_\theta(\mathbf{x}_q) \, \rangle. \tag{14}$$

Figure 3 compares the architectures of the Transformer and CoQE. The Transformer also contains token-wise components like feed-forward networks, and contextual components like multi-head attention modules. When stacked, these modules collectively exhibit contextual behavior, and the final token output intertwines with the context information in a complex manner during the forward pass. In contrast, CoQE explicitly separates the contextual and token-wise parts, which are responsible for learning the task representation space and the sample representation space, respectively. The two spaces interact through a well-defined inner product according to the Riesz representation theorem.

We aim to evaluate our model on regression tasks for ICL performance, and few-shot classification tasks for ICL-IWL performance. In the following, we will give the specific implementation of CoQE. Notably, due to their different properties, the task encoder constructs the task representation space in distinct ways.

### 6.1 Implementation for Regression

We employ a two-layer ReLU network as the sample encoder of CoQE, and a GPT-2–style Transformer as the task encoder. We take the final output token of the task encoder directly as the task representation induced by the context. The regression output is then computed as the inner product between it and the query sample representation. For fair comparison, the baseline Transformer is also equipped with the same two-layer ReLU embedding module. See Appendix B.2 for more implementation details.

### 6.2 Implementation for Few-shot Classification

We use a ResNet as CoQE's sample encoder, and a Transfromer as the task encoder. The task representation space is constructed in a distinct way from regression. A multi-class classification task can be regarded as a collection of sub-tasks that identify each class. Thus, we let it correspond to a set of task representations, each of which is associated with one

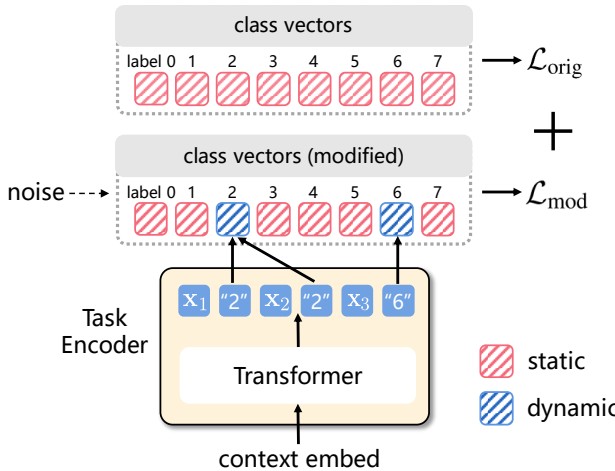

Figure 4: Construction and training of the task Representation space for few-shot classification.

class. ICL requires producing the task representations corresponding to the classes in the context, whereas IWL requires static memorization of all classes. To construct a task representation set compatible with both, we assign a parameterized vector to each class, representing a static version of its task representation.

In each forward pass, the classes appearing in the context are encoded by the task encoder to obtain their corresponding task representations, as illustrated in Figure 4. These dynamic vectors replace the corresponding static class vectors, and modified class vectors are used to compute logits for prediction. The resulting training loss is denoted as $\mathcal{L}_{\mathrm{mod}}$. Additionally, to accelerate the training of the static class vectors, we compute an additional set of logits from the unmodified class vectors during training, with the resulting classification loss denoted as $\mathcal{L}_{\mathrm{orig}}$. These logits are not used during testing. Therefore, the total training loss is $\mathcal{L}_{\mathrm{mod}} + \mathcal{L}_{\mathrm{orig}}$. During experiments, we observed that $\mathcal{L}_{\mathrm{mod}}$ tends to converge to $\mathcal{L}_{\mathrm{orig}}$, which means the task encoder fails to dynamically encode the context over training, and the learning of the task representation space is restricted to the set of static class vectors. To prevent the task representation space from collapsing during training, we add Gaussian noise $\epsilon$ to the modified logits during training, with the variance increasing over training steps. The initial noise follows $\mathcal{N}(\mu_0, 1)$. This trick can be interpreted as indirectly performing random sampling in the task representation space, that is, $\langle \omega_f, \phi(\mathbf{x}) \rangle + \epsilon = \langle \omega_f + \epsilon', \phi(\mathbf{x}) \rangle$. Through it we maintain a dynamic task representation space. See Section 7.2 for experimental results and ablation studies.

## 7 Experiments

In this section, we first evaluate the ICL performance of CoQE on regression tasks. Then we evaluate ICL-IWL performance on the synthetic classification task and our newly designed pseudo-arithmetic task.

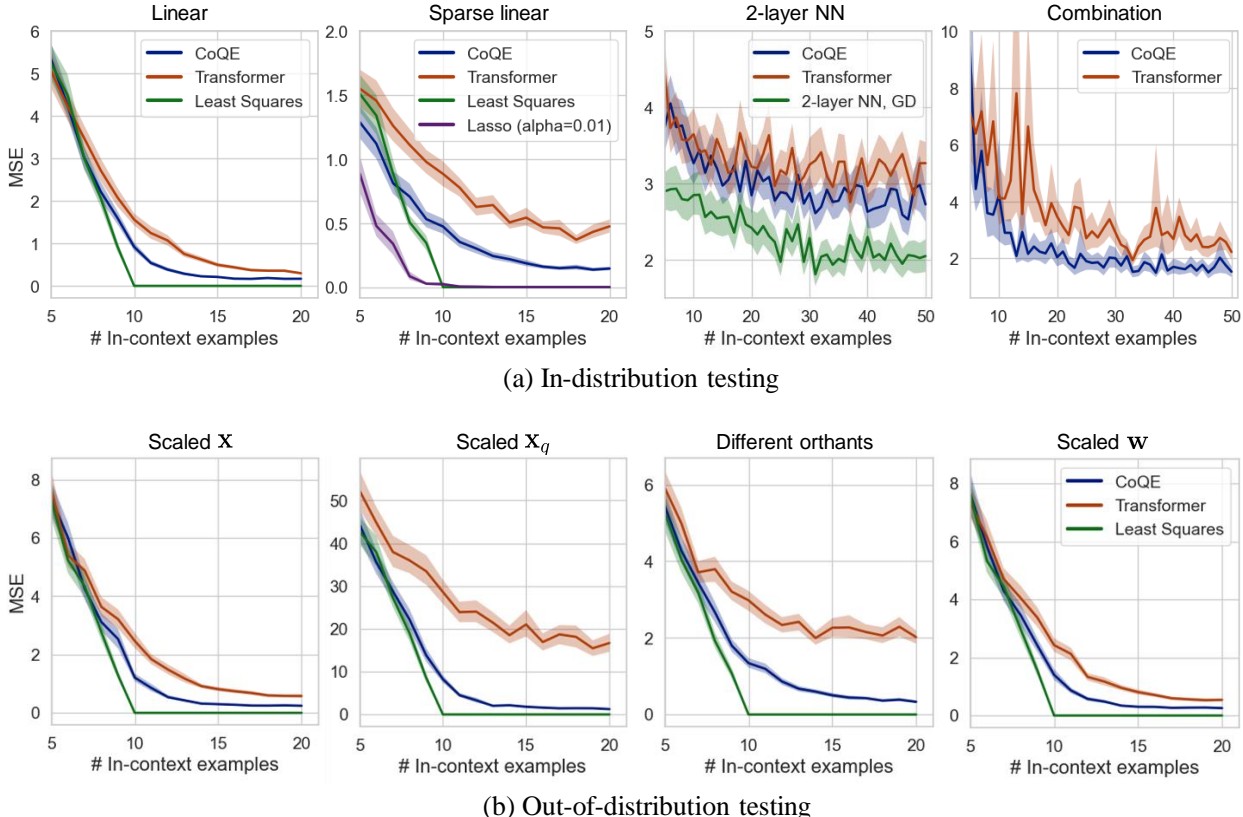

(a) In-distribution testing

(b) Out-of-distribution testing

Figure 5: Results of ICL regression. We provide optimal baselines for test settings except for combination functions. CoQE consistently achieves lower ICL error than the Transformer in both ID and OOD scenarios.

## 7.1 Regression

**Setup.** We adopt a general framework for training models to perform ICL over a function class $\mathcal{F}$. To construct training prompts, we first sample a task function $f \sim \mathcal{D}_{\mathcal{F}}^{\text{train}}$, then draw $k$ i.i.d. inputs $\mathbf{x}_1, \ldots, \mathbf{x}_k \sim \mathcal{D}_{\mathcal{X}}^{\text{train}}$. The prompt is formed as $\mathcal{P} = (\mathbf{x}_1, f(\mathbf{x}_1), \ldots, \mathbf{x}_k, f(\mathbf{x}_k))$. Let $\mathcal{P}^i$ denotes the prefix containing the first $i$ input-output examples and the $(i+1)$th input: $\mathcal{P}^i = (\mathbf{x}_1, f(\mathbf{x}_1), \ldots, \mathbf{x}_i, f(\mathbf{x}_i), \mathbf{x}_{i+1})$. The training objective of a model $\mathbb{M}_\theta$ minimizes the expected loss over all possible prefixes:

$$\min_\theta \ \mathbb{E}_{\mathcal{P}} \left[ \frac{1}{k} \sum_{i=0}^{k-1} \ell\big(\mathbb{M}_\theta(\mathcal{P}^i), \ f(\mathbf{x}_{i+1})\big) \right],$$

where $\ell(\cdot, \cdot)$ is a mean squared error (MSE) loss function. At test time, we first sample a test function $f \sim \mathcal{D}_{\mathcal{F}}^{\text{test}}$, then draw $j \leq k-1$ inputs $\mathbf{x}_1, \ldots, \mathbf{x}_j \sim \mathcal{D}_{\mathcal{X}}^{\text{test}}$, and $\mathbf{x}_q$ from $\mathcal{D}_{\text{query}}$ to construct the test prompt: $\mathcal{P}_{\text{test}}^j = (\mathbf{x}_1, f(\mathbf{x}_1), \ldots, \mathbf{x}_j, f(\mathbf{x}_j), \mathbf{x}_q)$. Then we measure the MSE between $\mathbb{M}_\theta(\mathcal{P}_{\text{test}}^j)$ and $f(\mathbf{x}_q)$.

To compare ICL capability of CoQE with the standard Transformer, we consider two major evaluation scenarios: in-distribution (ID) testing and out-of-distribution (OOD) testing. For ID testing, we set $\mathcal{D}_{\mathcal{X}}^{\text{train}} = \mathcal{D}_{\mathcal{X}}^{\text{test}} = \mathcal{D}_{\text{query}}$, and $\mathcal{D}_{\mathcal{F}}^{\text{train}} = \mathcal{D}_{\mathcal{F}}^{\text{test}}$. Specifically, we use the following four classes of functions $\mathcal{F}$: linear functions, sparse linear functions, two-layer ReLU networks and combination functions. The latter two classes of nonlinear functions allow the model to reduce ICL difficulty by learning task-invariant representations. Through them, we can empirically validate Theorem 5.6, which shows the benefits of dual-space modeling for representation learning. For OOD testing, we consider four different cases of distribution shifts under linear functions. See Appendix B.2 for more setup details.

**Results.** In the ID scenario, CoQE consistently achieves lower ICL error than the Transformer (Figure 5 (a)). For regression on more challenging combination functions, the Transformer exhibits substantial fluctuations, whereas CoQE attains much smaller error variance. We attribute this to CoQE's more effective

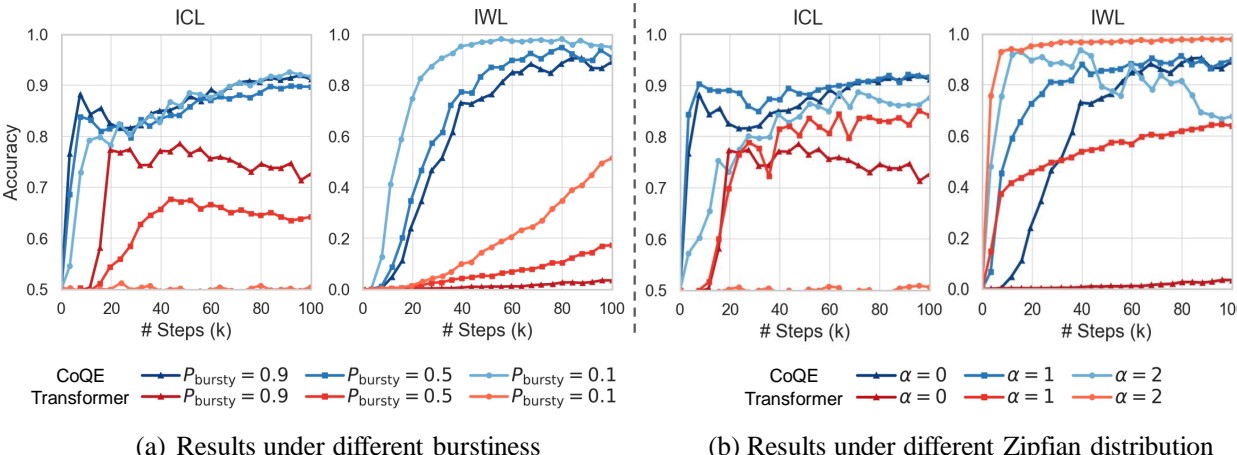

(a) Results under different burstiness        (b) Results under different Zipfian distribution

Figure 6: Learning curves of synthetic few-shot classification under different data distribution factors.

learning of the sample representation space, and present further results in Appendix B.2. In the OOD scenario, CoQE also achieves substantially lower error than the Transformer across all four tested cases (Figure 5 (b)). Notably, the second case is adapted from Mittal et al. (2025), who similarly aims to enforce the model to explicitly learn task variables. However, they found no improvement in OOD performance, contrary to our results. This indicates that simply introducing task variables is insufficient and highlights the value of our proposed dual-space modeling and corresponding architecture design.

## 7.2 Synthetic Few-shot Classification

**Baselines.** Besides standard Transformers, we introduce two more baselines that were proposed to alleviate the tension between ICL and IWL: $L_2$ regularization (Singh et al., 2023) and probabilistic temporary forgetting (Anand et al., 2025).

**Results.** Figure 1 top right presents the ICL and IWL accuracies of Transformers and CoQE under various factors after 100k training steps. The Transformers fluctuate between ICL and IWL capabilities across different conditions, whereas our models robustly occupy the upper-right blue region, indicating a Pareto improvement in both abilities. The other two baselines perform even worse than the standard Transformer; for clarity, we omit them from the figure, and report the detailed results in Appendix B.3. Figure 6 shows the learning curves under different values of $P_{bursty}$ and Zipfian exponent. We could observe that CoQE's ICL accuracy rises rapidly at the beginning, but declines slightly between 10k and 30k steps. This behavior aligns with prior findings on ICL strategy: it emerges quickly and then gradually fades (Singh et al., 2023). However, under our algorithm, the model quickly restrains this fading trend and continues to recover steadily. We also discuss the effect of parameter scale, showing that CoQE outperforms the standard Transformer under the same parameter budget, and its performance continues to improve as the model size increases. All detailed results are presented in Appendix B.3.

**Ablation study.** We study the effect of Gaussian noise on the model performance, as shown in Table 1. Without any noise, the model's ICL ability ultimately yields entirely to IWL. When $\mu_0 = 5$, the model achieves maximal ICL performance while retaining high IWL capability. This is the default noise magnitude used in our experiments. See Appendix B.3 for more analysis and details.

## 7.3 Extending to Language Model Token Embeddings

To further verify the generality of our approach and explore its potential extension to natural language processing (NLP) tasks, we are inspired by Singh et al. (2023) and extend our few-shot classification experiments from images to token embeddings of large language models (LLMs). we construct a fixed token embedding dataset through selection and clustering for token embedding matrix of Llama series (Dubey et al., 2024), see Appendix B.4 for details. Compared with Omniglot images, this dataset has larger intra-class variability and preserves meaningful semantic information relevant to NLP tasks. We fix the training data distribution

Table 1: Results under different noises. We use $\mu_0 = 5$ as default noise magnitude.

|  | ICL | IWL |
|---|---|---|
| Noise-free | 55.12 | 99.62 |
| $\mu_0 = 3$ | 81.91 | 95.31 |
| $\mu_0 = 5$ | 91.15 | 89.30 |
| $\mu_0 = 7$ | 88.22 | 77.70 |
| $\mu_0 = 9$ | 86.01 | 72.62 |

Table 2: Results of using language token embeddings.

| Token Embedding | Model | ICL | IWL |
|---|---|---|---|
| Llama-3.1-8B | Transformer | 50.12 | 96.65 |
|  | CoQE | 65.35 | 93.96 |
| Llama-3.2-3B | Transformer | 49.33 | 98.81 |
|  | CoQE | 75.01 | 96.57 |
| Llama-3.2-1B | Transformer | 49.95 | 99.12 |
|  | CoQE | 80.08 | 97.34 |

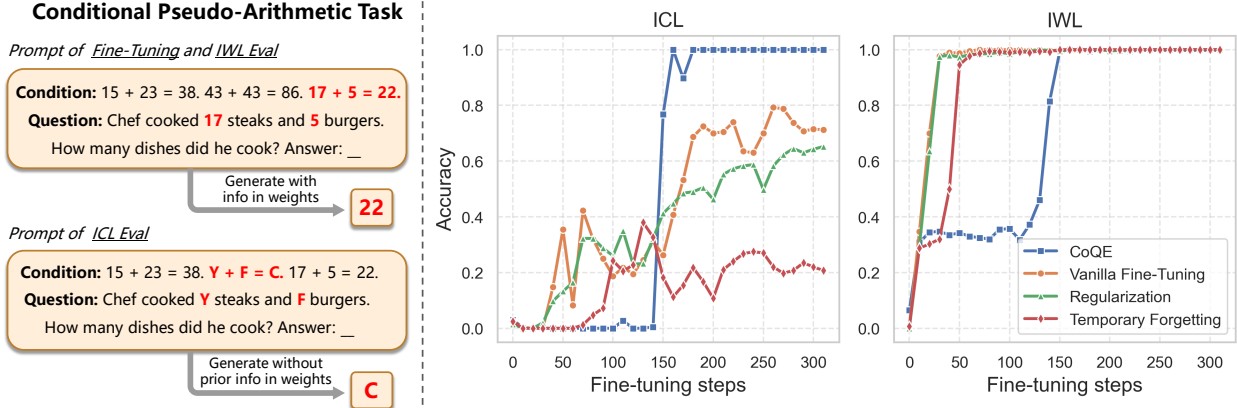

Figure 7: (Left) Overview of our conditional pseudo-arithmetic task. (Right) Training curves of CoQE and three baselines. The IWL accuracy of CoQE improves more slowly, because its learning paradigm differs from the model's original autoregressive pre-training. However, after fine-tuning, all methods achieve nearly 100% IWL performance, and CoQE achieves substantially higher ICL accuracy.

parameters at $P_{\mathrm{bursty}} = 0.9$ and Zipfian exponent $\alpha = 1$, to approximate natural language distributions. The results in Table 2 show that the Transformer almost fails to acquire ICL ability, consistent with Singh et al. (2023). In contrast, CoQE achieves substantially better ICL capability while maintaining strong IWL performance. We believe these results demonstrate that our dual spaces encoding can enhance ICL capability and mitigate the ICL–IWL conflict in NLP scenarios where semantic information is richer.

## 7.4 Conditional Pseudo-Arithmetic Task

To further verify the generalization ability of our algorithm on generative tasks and pretrained models, we designed a conditional pseudo-arithmetic task, inspired by Anand et al. (2025) and Chen et al. (2025). We construct sequences consisting of arithmetic conditions followed by a word problem, as shown in Figure 7. Here, the question can be answered solely based on the given conditions. We fine-tune GPT-2 on 5,000 such sequences and evaluate its accuracy on the same distribution, which we interpret as its IWL performance. After fine-tuning, we then construct sequences that mix standard arithmetic with "pseudo-arithmetic" over letters. In this case, the model must rely on the contextual relation given in the sequence (rather than memorized weights) to answer correctly, which we use to measure ICL performance. In this setup, $\mathbf{x}_i$ and $\mathbf{x}_q$ are generalized from single tokens to sequence-level inputs.

We extend CoQE to the pretrained GPT-2 as follows. During fine-tuning, we modify the attention mask so that tokens in the Question field cannot attend to the Condition field, thereby implementing separate Context–Query Encoding. On the output side, we largely follow the implementation for few-shot classification. The difference is that we do not introduce $\mathcal{L}_{\mathrm{orig}}$ or noise injection, because we observe that fine-tuning pretrained GPT-2 on this dataset is fast and stable. Besides standard fine-tuning, we also introduce $L_2$

regularization and probabilistic temporary forgetting as baselines. All models are fine-tuned for 300 steps. The training curves are shown in Figure 7 right.

After fine-tuning, all methods achieve nearly 100% IWL performance, while CoQE achieves substantially higher ICL accuracy than the other three baselines. A typical failure mode we observe is using the wrong condition, for example: "Condition: $42 + 4 = 46$. $31 + 20 = 51$. $L + Y = E$. Question: I read L pages yesterday and Y pages today. How many pages did I read? Answer: 51." This indicates that the model is still limited in leveraging the correct in-context relation in this scenario. Moreover, $L_2$ regularization and probabilistic temporary forgetting perform even worse than vanilla fine-tuning. This demonstrate limitations of their design: regularization can take effect at very large training steps (e.g., $10^7$), and probabilistic temporary forgetting targets ICL settings that involve random-vector inputs, which differs from our pseudo-arithmetic setting. Overall, this simple autoregressive experiment further validates the effectiveness and transferability of CoQE on generative tasks and pretrained models.

## 8 Discussion

Our analyses and experiments primarily focus on small models. A natural question is whether large models exhibit similar imbalance between ICL and IWL. Piantadosi (2014) showed that a Zipfian distribution of $\alpha = 1$ closely approximates the empirical distribution of natural language, which serves as a sweet spot for the tradeoff between ICL and IWL (Chan et al., 2022). Nevertheless, Chan et al. (2025) pointed out that LLMs can still experience conflicts between ICL and IWL when demonstration examples deviate from the training distribution. Beyond this, the growing demand for multimodal large models (e.g., VLMs, VLAs) trained on increasingly diverse data distributions, as well as the emergence of new architectures, makes it less tenable to rely on the fortunate coincidence of natural language statistics to balance ICL and IWL. These considerations underscore the importance of our exploration in this direction.

**Limitations.** Theoretically, a limitation of our work is that our modeling is built upon the linear representation hypothesis. Although this assumption also underlies many other theoretical and empirical studies, its applicability to complex NLP tasks like multi-step reasoning still requires further evidence. Another limitation is that our dual-space formulation currently assumes that the answer $y_q$ is a single value rather than a sequence. Extending the theoretical modeling to handle open-ended generation tasks is valuable.

Emperically, the main limitation is that the evaluation of our algorithm has so far been confined to structural synthetic experiments. For tasks where such a context–query decomposition is not available or not well defined, our current algorithm does not yet provide an effective way to apply dual-space modeling. Extending our algorithm to broader general NLP or multimodal settings is also an important direction for future work.

### Broader Impact Statement

Reconciling ICL and IWL under diverse conditions is crucial for the robustness of large models as data distributions grow more varied and architectures evolve. This work aims to enhance generalization and reliability of large models, contributing to safer and more equitable deployment in real-world applications.

### Acknowledgments

This work was supported in part by the National Key Research and Development Program of China (No. 2024YDLN0006), in part by the National Key Research and Development Program of China under STI 2030—Major Projects (No. 2021ZD0200300), and in part by the Tsinghua-Fuzhou Data Technology Joint Research Institute (Project No. JIDT2024013).

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

# A  Proofs of Theoretical Results

## A.1  Proof of Proposition 5.3

For ease of presentation, we first restate the proposition and then introduce its proof.

**Proposition A.1** (Task-sample duality)**.** *Let $\mathcal{X}$ be the input space and $\mathcal{Y}_f$ the multiple label sets corresponding to each task $f \in \mathcal{F}$. Under Definition 5.2, there exists a linear sample representation space $\mathcal{M}_\mathcal{F}$ and a linear task transformation space $\mathcal{T}$, where $\mathcal{T}$ is the dual space of $\mathcal{M}_\mathcal{F}$, i.e. $\mathcal{T} = \mathcal{M}_\mathcal{F}^*$.*

*Proof.* To prove the proposition, we must show that the linear task transformation space $\mathcal{T}$ is equivalent to the dual space of the linear sample representation space $\mathcal{M}_\mathcal{F}$, denoted as $\mathcal{M}_\mathcal{F}^*$. The proof proceeds by demonstrating mutual inclusion: (1) $\mathcal{T} \subseteq \mathcal{M}_\mathcal{F}^*$ and (2) $\mathcal{M}_\mathcal{F}^* \subseteq \mathcal{T}$.

**Step 1: Proof of $\mathcal{T} \subseteq \mathcal{M}_\mathcal{F}^*$.**  Let $t$ be an arbitrary element in the task transformation space $\mathcal{T}$. According to Definition 5.2, $t$ is a linear function such that

$$t(m) = \langle \omega_t, m \rangle, \quad \forall m \in \mathcal{M}_\mathcal{F}.$$

Since $t$ is a linear functional on $M_\mathcal{F}$, it is by definition an element of $M_\mathcal{F}^*$. As $t$ was an arbitrary element of $\mathcal{T}$, it follows that every element in $\mathcal{T}$ corresponds to a unique linear functional in $M_\mathcal{F}^*$. Thus, we have established that $\mathcal{T} \subseteq \mathcal{M}_\mathcal{F}^*$.

**Step 2: Proof of $\mathcal{M}_\mathcal{F}^* \subseteq \mathcal{T}$.**  Conversely, let $t'$ be an arbitrary linear functional in the dual space $\mathcal{M}_\mathcal{F}^*$. By Definition 5.1, $\mathcal{M}_\mathcal{F}$ is a finite-dimensional inner product space. By the Riesz representation theorem, for any linear functional $t' \in \mathcal{M}_\mathcal{F}^*$, there exists a unique vector, let's call it $\omega_{t'} \in \mathcal{M}_\mathcal{F}$, such that for all $m \in \mathcal{M}_\mathcal{F}$:

$$t'(m) = \langle \omega_{t'}, m \rangle.$$

Now, let us define a function $f_{t'} : \mathcal{X} \to \mathbb{R}$ using this functional $t'$:

$$f_{t'}(\mathbf{x}) = t'(\phi_\mathcal{F}(\mathbf{x})) = \langle \omega_{t'}, \phi_\mathcal{F}(\mathbf{x}) \rangle.$$

This function $f_{t'}$ has the exact mathematical form of a task function as specified in Definition 5.2. Therefore, $f_{t'}$ can be considered a valid task belonging to the task function space $\mathcal{F}$. Definition 5.2 states that for any such task $f_{t'} \in \mathcal{F}$, there exists a unique linear task representation vector, which we denote $\omega_f$, that represents it. This means:

$$f_{t'}(\mathbf{x}) = \langle \omega_f, \phi_\mathcal{F}(\mathbf{x}) \rangle.$$

By equating the two expressions for $f_{t'}(\mathbf{x})$, we obtain:

$$\langle \omega_{t'}, \phi_\mathcal{F}(\mathbf{x}) \rangle = \langle \omega_f, \phi_\mathcal{F}(\mathbf{x}) \rangle, \quad \forall \mathbf{x} \in \mathcal{X}$$

This implies that $\langle \omega_{t'} - \omega_f, m \rangle = 0$ for all $m$ in the image of $\phi_\mathcal{F}$. Since the sample representation space $\mathcal{M}_\mathcal{F}$ is spanned by the image of $\phi_\mathcal{F}$, this condition holds for all $m \in \mathcal{M}_\mathcal{F}$. The only vector orthogonal to every vector in an inner product space is the zero vector. Therefore:

$$\langle \omega_{t'} - \omega_f, m \rangle = 0 \implies \omega_{t'} = \omega_f.$$

Since $\omega_f$ corresponds to an element of $\mathcal{T}$, it follows that $\omega_{t'}$ is also corresponds to an element of $\mathcal{T}$, and thus $t' \in \mathcal{T}$. As our choice of $t'$ was arbitrary, we have shown that every linear functional in $\mathcal{M}_\mathcal{F}^*$ corresponds to an element in $\mathcal{T}$. Thus, we have established that $\mathcal{M}_\mathcal{F}^* \subseteq \mathcal{T}$. $\square$

## A.2  Proof of Theorem 5.6

For ease of presentation, we first restate the theorem and then introduce its proof.

**Theorem A.2** (Completeness of basis representations under task traversal). *Under Proposition 5.3, we assume that a learner with sample representation mapping $\phi_\theta$ is presented with a task traversal curriculum $\mathcal{C}$ such that:* $\text{span}\{t \mid t \in \mathcal{C}\} = \mathcal{T}$. *Then, if the learner achieves zero empirical error, the learned representation mapping $\phi_\theta$ satisfies:* $\text{span}\{\phi_\theta(\mathbf{x}) \mid \mathbf{x} \in \mathcal{X}\} = \mathcal{M}_\mathcal{F}$.

*Proof.* By Proposition 5.3, fix a basis $\{m_i\}_{i=1}^d$ of $\mathcal{M}_\mathcal{F}$ and its dual basis $\{t_i\}_{i=1}^d \subset \mathcal{T}$, satisfying $t_i(m_j) = \langle \omega_i, m_j \rangle = \delta_{ij}$. For any $m \in \mathcal{M}_\mathcal{F}$ write the unique decomposition $m = \sum_{i=1}^d \alpha_i(m) \, m_i$ and for any $t \in \mathcal{T}$ write $t = \sum_{i=1}^d \beta_i(t) \, t_i$. The bilinear pairing then reduces to

$$t(m) = \sum_{i=1}^d \alpha_i(m) \, \beta_i(t). \tag{15}$$

Let $\phi_\mathcal{F} : \mathcal{X} \to \mathcal{M}_\mathcal{F}$ denote the sample representation guaranteed by Definition 5.2, and define the coordinate vectors

$$\alpha^\theta(\mathbf{x}) \triangleq (\alpha_1(\phi_\theta(\mathbf{x})), \ldots, \alpha_d(\phi_\theta(\mathbf{x}))) \in \mathbb{R}^d, \qquad \alpha^*(\mathbf{x}) \triangleq (\alpha_1(\phi_\mathcal{F}(\mathbf{x})), \ldots, \alpha_d(\phi_\mathcal{F}(\mathbf{x}))) \in \mathbb{R}^d.$$

Zero empirical error on every curriculum task $t \in \mathcal{C}$ means

$$t(\phi_\theta(\mathbf{x})) = t(\phi_\mathcal{F}(\mathbf{x})) \quad \text{for all } t \in \mathcal{C} \text{ and all training } \mathbf{x}.$$

By linearity of Equation 15 this equality holds for any linear combination of curriculum tasks; hence it holds for all $t \in \text{span}(\mathcal{C}) = \mathcal{T}$:

$$t(\phi_\theta(\mathbf{x})) = t(\phi_\mathcal{F}(\mathbf{x})), \qquad \forall \, t \in \mathcal{T}. \tag{16}$$

Take in Equation 16 the particular choice $t = t_i$ (the $i$-th dual basis functional). Using $t_i(m) = \alpha_i(m)$ from Equation 15, we obtain for every $\mathbf{x}$ and every $i \in [d]$,

$$\alpha_i(\phi_\theta(\mathbf{x})) = t_i(\phi_\theta(\mathbf{x})) = t_i(\phi_\mathcal{F}(\mathbf{x})) = \alpha_i(\phi_\mathcal{F}(\mathbf{x})).$$

Thus $\alpha^\theta(\mathbf{x}) = \alpha^*(\mathbf{x})$ pointwise for all (training) $\mathbf{x}$. Consequently $\phi_\theta(\mathbf{x})$ and $\phi_\mathcal{F}(\mathbf{x})$ have identical coordinates in the basis $\{m_i\}_{i=1}^d$ for all $\mathbf{x}$, so

$$\text{span}\{\phi_\theta(\mathbf{x}) \mid \mathbf{x} \in \mathcal{X}\} = \text{span}\{\phi_\mathcal{F}(\mathbf{x}) \mid \mathbf{x} \in \mathcal{X}\}.$$

Without loss of generality, take $\mathcal{M}_\mathcal{F} = \text{span}\{\phi_\mathcal{F}(\mathbf{x}) \mid \mathbf{x} \in \mathcal{X}\}$. Therefore $\text{span}\{\phi_\theta(\mathbf{x}) \mid \mathbf{x} \in \mathcal{X}\} = \mathcal{M}_\mathcal{F}$, proving the claim.

$\square$

## A.3 Proof of Proposition 5.8

For ease of presentation, we first restate the proposition and then introduce its proof.

**Proposition A.3** (Closed form of $\omega_f$ under simplified LSA). *Consider an LSA layer applied after a feature encoder $\phi : \mathcal{X} \to \mathbb{R}^d$ implemented by an MLP. Suppose the LSA projection matrices $W_{KQ}$ and $W_{OA}$ are initialized such that*

$$W_{OV} = \begin{pmatrix} * & * \\ 0_d^\top & 1 \end{pmatrix}, \qquad W_{KQ} = \begin{pmatrix} \Theta & 0_d \\ 0_d^\top & * \end{pmatrix}.$$

*Then the final prediction takes the form $\hat{y} = \langle \omega_f(\mathbf{z}_{1:n}, \phi), \phi(\mathbf{x}_q) \rangle$, where*

$$\omega_f(\mathbf{z}_{1:n}, \phi) = \frac{1}{n} \sum_{i=1}^n \mathbf{y}_i \Theta^\top \phi(\mathbf{x}_i). \tag{17}$$

*Proof.* According to Kim & Suzuki (2024), under the conditions of Proposition 5.8, the expression of $\hat{y}$ is given as follows:

$$\hat{y} = \frac{1}{n} \sum_{i=1}^n \mathbf{y}_i \phi(\mathbf{x}_i)^\top \Theta \phi(\mathbf{x}_q). \tag{18}$$

Hence, Proposition 5.8 is readily proved.

$\square$

## A.4 Proof of Theorem 5.9

For ease of presentation, we first restate the theorem and then introduce its proof.

**Theorem A.4** (Entangled structure under general SA). *For a standard SA model with softmax-based attention weights, there does NOT exist a pair of $\phi_0$ and $\omega_0(\mathbf{z}_{1:n}, \phi_0)$, such that the model prediction admits the following decomposition:*

$$\hat{\mathbf{y}}_q = \langle \omega_0(\mathbf{z}_{1:n}, \phi_0), \ \phi_0(\mathbf{x}_q) \rangle. \tag{19}$$

*Proof.* We argue by contradiction. Assume there exists a finite-dimensional feature map $\phi_0$ and a context-only coefficient vector $\omega_0(\mathbf{z}_{1:n}, \phi_0)$ such that the identity holds for all contexts and queries.

**Step 1: From SA equations to a ratio of exponentials in $\mathbf{x}_q$.** Let the sequence length be $L = n+1$. Stack token embeddings as $Z = [\mathbf{z}_1, \dots, \mathbf{z}_n, \mathbf{z}_q] \in \mathbb{R}^{d \times L}$. A single-head self-attention (SA) layer computes

$$Q = W_Q Z, \quad K = W_K Z, \quad V = W_V Z,$$

with $Q, K \in \mathbb{R}^{d_k \times L}$, $V \in \mathbb{R}^{d_v \times L}$. Denote the $i$-th key/value columns by $k_i := K_{:i} = W_K \mathbf{z}_i$, $v_i := V_{:i} = W_V \mathbf{z}_i$, and the query column at position $q$ by $q := Q_{:L} = W_Q \mathbf{z}_q$. The attention weights for the query position form a probability vector $\alpha(q) \in \Delta^n$ with coordinates

$$\alpha_i(q) \ = \ \frac{\exp\left(\langle k_i, q \rangle / \sqrt{d_k}\right)}{\sum_{j=1}^{L} \exp\left(\langle k_j, q \rangle / \sqrt{d_k}\right)}, \qquad i = 1, \dots, L. \tag{20}$$

In the theoretical analysis of ICL, it is common to set $\mathbf{z}_q = [\mathbf{x}_q, 0]$. Without loss of generality, we assume that the query token embedding depends affinely on the input feature $\mathbf{x}_q \in \mathbb{R}^{d_x}$:

$$\mathbf{z}_q \ = \ E_x \mathbf{x}_q + r_q,$$

where $E_x \in \mathbb{R}^{d \times d_x}$ is a fixed embedding matrix and $r_q \in \mathbb{R}^d$ could collect position encodings and other context-independent parts at position $q$. Then the query vector is also affine in $\mathbf{x}_q$:

$$q \ = \ W_Q \mathbf{z}_q \ = \ W_Q E_x \mathbf{x}_q + W_Q r_q \ = \ U \mathbf{x}_q + u_0,$$

with $U := W_Q E_x \in \mathbb{R}^{d_k \times d_x}$ and $u_0 := W_Q r_q \in \mathbb{R}^{d_k}$. Plugging $q = U \mathbf{x}_q + u_0$ into the logits in Equation 20 yields, for each key $i$,

$$\frac{\langle k_i, q \rangle}{\sqrt{d_k}} \ = \ \frac{\langle k_i, U \mathbf{x}_q \rangle}{\sqrt{d_k}} + \frac{\langle k_i, u_0 \rangle}{\sqrt{d_k}} = a_i^\top \mathbf{x}_q + b_i(\mathbf{z}),$$

where we define the (query–input) slope and the (context) offset by

$$a_i \ := \ \frac{U^\top k_i}{\sqrt{d_k}} \in \mathbb{R}^{d_x}, \qquad b_i(\mathbf{z}) \ := \ \frac{\langle k_i, u_0 \rangle}{\sqrt{d_k}} \in \mathbb{R}.$$

Hence, for a fixed context $\mathbf{z}_{1:n}$ (which fixes all $k_i$ and $u_0$), the attention weights are *softmax of affine functions of $\mathbf{x}_q$*:

$$\alpha_i(\mathbf{x}_q; \mathbf{z}) \ = \ \frac{\exp\left(a_i^\top \mathbf{x}_q + b_i(\mathbf{z})\right)}{\sum_{j=1}^{L} \exp\left(a_j^\top \mathbf{x}_q + b_j(\mathbf{z})\right)}, \qquad i = 1, \dots, L. \tag{21}$$

The SA output at the query position is $h_q \ = \ \mathbf{z}_q + W_O \sum_{i=1}^{L} \alpha_i(\mathbf{x}_q; \mathbf{z}) v_i$. For a fixed linear predictor $w \in \mathbb{R}^d$ (or equivalently choosing a fixed output coordinate), the scalar prediction is

$$\hat{\mathbf{y}}_q(\mathbf{x}_q) = w^\top h_q = \underbrace{w^\top \mathbf{z}_q}_{\text{affine in } \mathbf{x}_q} + \sum_{i=1}^{L} \underbrace{\left(w^\top W_O v_i\right)}_{:= \gamma_i(\mathbf{z})} \alpha_i(\mathbf{x}_q; \mathbf{z}). \tag{22}$$

If we choose $w$ orthogonal to $\mathrm{Im}(E_x)$ (always possible unless $E_x = 0$), then $w^\top \mathbf{z}_q = w^\top (E_x \mathbf{x}_q + r_q) = w^\top r_q$ is a context-only constant; denote $c(\mathbf{z}) := w^\top r_q$. With $\gamma(\mathbf{z}) := (\gamma_1(\mathbf{z}), \dots, \gamma_L(\mathbf{z}))^\top$, Equation 22 simplifies to

$$\hat{\mathbf{y}}_q(\mathbf{x}_q) \ = \ c(\mathbf{z}) \ + \ \gamma(\mathbf{z})^\top \alpha(\mathbf{x}_q; \mathbf{z}), \tag{23}$$

where $\alpha(\cdot; \mathbf{z})$ is given by the ratio-of-exponentials form in Equation 21. This exhibits the claimed dependence of $\hat{\mathbf{y}}_q$ on $\mathbf{x}_q$ through a softmax over affine functions of $\mathbf{x}_q$.

**Step 2: A two-key reduction yields a linearly independent logistic family.** Specialize to $d_x = 1$ and one context keys ($n = 1$) with $a_1 \neq a_2$. Choose $W_O, V$ so that $c(z) \equiv 0$ and $\gamma_1(z) = 1, \gamma_2(z) = 0$. Then Equation 23 reduces to

$$\hat{y}_q(x_q) \;=\; \frac{\exp(a_1 x_q + b_1(z))}{\exp(a_1 x_q + b_1(z)) + \exp(a_2 x_q + b_2(z))} \;=\; \frac{1}{1 + t(z)\, e^{-a x_q}},$$

where $a := a_1 - a_2 \neq 0$ and $t(z) := \exp\big(b_2(z) - b_1(z)\big) > 0$. As the context varies, $t(z)$ can take arbitrarily many distinct positive values, so SA realizes the one-parameter family of functions

$$\mathcal{F} \;=\; \Big\{ f_t(x) := \frac{1}{1 + te^{-ax}} \;:\; t > 0 \Big\}.$$

Fix distinct $t_1, \ldots, t_m > 0$. Suppose there exist scalars $\lambda_1, \ldots, \lambda_m$ with $\sum_{i=1}^m \lambda_i f_{t_i}(x) \equiv 0$ for all $x \in \mathbb{R}$. Multiplying both sides by $\prod_{i=1}^m (1 + t_i e^{-ax})$ and letting $s = e^{-ax}$ gives the polynomial identity

$$\sum_{i=1}^m \lambda_i \prod_{j \neq i} (1 + t_j s) \;\equiv\; 0 \quad \text{for all } s > 0.$$

A polynomial that vanishes on an infinite set is identically zero; hence the identity holds for all $s \in \mathbb{R}$. Evaluating at $s = -1/t_k$ yields

$$\lambda_k \prod_{j \neq k} \left( 1 - \frac{t_j}{t_k} \right) \;=\; 0.$$

Since the $t_i$ are distinct, each product is nonzero, forcing $\lambda_k = 0$ for all $k$. Thus $f_{t_1}, \ldots, f_{t_m}$ are linearly independent. Consequently, the linear span of $\mathcal{F}$ is infinite-dimensional.

**Step 3: Contradiction with any finite-dimensional bilinear decomposition.** If the bilinear decomposition $\hat{y}_q(x_q) = \langle \omega_0(z), \phi_0(x_q) \rangle$ held with a *fixed* feature map $\phi_0 : \mathbb{R} \to \mathbb{R}^d$ (independent of the context), then for all contexts the functions $x_q \mapsto \hat{y}_q(x_q)$ would lie in the $d$-dimensional linear span of the coordinate functions of $\phi_0$. However, Step 2 shows that by varying the context, SA realizes an infinite set $\mathcal{F}$ of pairwise linearly independent functions in x, which cannot be contained in any finite-dimensional linear subspace. This contradiction rules out the existence of such $(\phi_0, \omega_0)$. □

## B Experimental Details and Additional Results

In this part of the appendix, we provide detailed descriptions of the experiments in the main text and include additional experimental results.

### B.1 Representation Visualization

To further illustrate the training dynamics in Figure 2 left, we visualize the evolution of representations for three common classes during training under training settings $(64, 12, 0.9, 2)$ and $(64, 12, 0.9, 0)$, as shown in Figure 8. We find that under different data distributions, the model's representation space diverges significantly from the very beginning of training and eventually converges to distinct structure. This clearly reflects that the balance of sample-level and context-level information is highly sensitive to the training distribution.

### B.2 Regression

**Implementation details.** We use Transformer architectures from the GPT-2 family (Radford et al., 2018) as implemented by HuggingFace (Wolf et al., 2020). Specifically, the Transformer baseline we use is configured with an embedding dimension of 64, 3 layers, and 2 attention heads, resulting in a total of 0.2M parameters. The task encoder of CoQE uses the exact same Transformer configuration. The representation encoder of CoQE consists of a two-layer ReLU network, implemented as a linear projection, followed by a

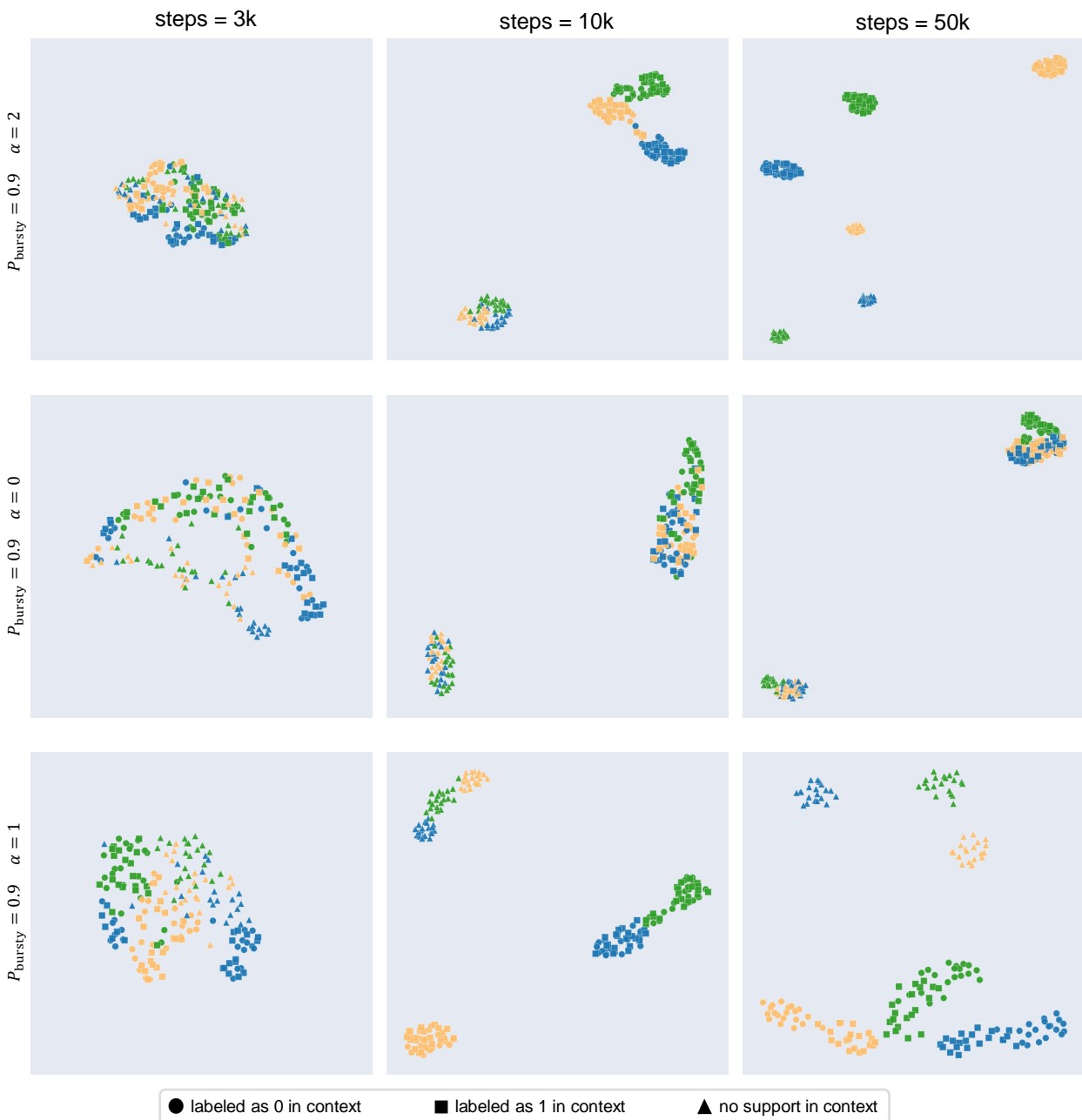

Figure 8: The uncropped visualization results of training dynamics.

ReLU activation, a LayerNorm, and a second linear layer. For fair comparison, the baseline Transformer's embedding module uses the exact same two-layer ReLU network. During training across the four classes of functions, we use a batch size of 64 and a learning rate of $5e-5$. For the three tasks except combination functions, models are trained for $1 \times 10^5$ steps, while the combination task is trained for $2 \times 10^5$ steps due to its increased difficulty. All experiments are conducted on an NVIDIA RTX 4090 GPU.

**Setup details.** We consider two major evaluation scenarios for regression: in-distribution (ID) testing and out-of-distribution (OOD) testing. In the ID scenario, we set $\mathcal{D}_{\mathcal{X}}^{\text{train}} = \mathcal{D}_{\mathcal{X}}^{\text{test}} = \mathcal{D}_{\text{query}}$, and $\mathcal{D}_{\mathcal{F}}^{\text{train}} = \mathcal{D}_{\mathcal{F}}^{\text{test}}$. Specifically, we use the following four classes of functions $\mathcal{F}$:

- Linear functions: $\mathcal{F} = \{f \mid f(\mathbf{x}) = \mathbf{w}^\top \mathbf{x}, \ \mathbf{w} \in \mathbb{R}^d\}$, where $d = 10$. We sample $\mathbf{x}_1, \ldots, \mathbf{x}_j, \mathbf{x}_q$ and $\mathbf{w}$ independently from the isotropic Gaussian distribution $\mathcal{N}(0, I_d)$, then compute $f(x_i) = \mathbf{w}^\top \mathbf{x}_i$ to construct the prompt. In this setting we use the least squares estimator as the optimal baseline.

- Sparse linear functions: $\mathcal{F} = \{f \mid f(\mathbf{x}) = \mathbf{w}^\top \mathbf{x}, \ \mathbf{w} \in \mathbb{R}^d, \|\mathbf{w}\|_0 \le s\}$, where $d = 10$ and $s = 3$. We also sample $\mathbf{x}_1, \ldots, \mathbf{x}_j, \mathbf{x}_q$ and $\mathbf{w}$ independently from $\mathcal{N}(0, I_d)$, and then zero out all but $s$ coordinates of $\mathbf{w}$ uniformly at random. We use the least squares estimator and Lasso, which leverages sparsity with an $\ell_1$-norm regularizer as baselines.

- Two-layer ReLU neural networks: $\mathcal{F} = \{f \mid f(\mathbf{x}) = \sum_{i=1}^{h} a_i \sigma(\mathbf{w}_i^\top \mathbf{x}), \ a_i \in \mathbb{R}, \ \mathbf{w}_i \in \mathbb{R}^d\}$, where $\sigma(z) = \max\{0, z\}$ is the ReLU activation function, and $d = 5$, $h = 10$. We sample $\mathbf{x}_i$s and $\mathbf{x}_q$ from $\mathcal{N}(0, I_d)$, along with network parameters $a_i$s from $\mathcal{N}(0, 2/h)$. We sample $\mathbf{w}_i$s from $\mathcal{N}(0, I_d)$, and share them across all tasks in $\mathcal{F}$. The baseline is a two-layer neural network of the same architecture trained on in-context examples using Adam.

- Combination functions: $\mathcal{F} = \{f \mid f(\mathbf{x}) = \mathbf{w}^\top \Phi(\mathbf{x}), \ \mathbf{w} \in \mathbb{R}^5\}$, where $\Phi$ is an element-wise combination function. For $\mathbf{x} = [\, x_1, x_2, x_3, x_4, x_5 \,]$, $\Phi(\mathbf{x}) = [\, |x_1|, \ x_2^2, \ x_3^3, \ \cos(\pi x_4), \ e^{0.2\, x_5} \,]^\top$. We sample $\mathbf{x}_i$s, $\mathbf{x}_q$ and $\mathbf{w}$ from $\mathcal{N}(0, I_5)$ independently. In this setting, there is no naturally optimal baseline, so we compare only with the Transformer.

The latter two classes of nonlinear functions allow the model to reduce ICL difficulty through representation learning, by learning task-invariant $\mathbf{w}_i$s or $\Phi$.

In the OOD scenario, we consider four different cases of distributional shifts under linear functions.

- $\mathcal{D}_{\mathcal{X}}^{\text{train}} \ne \mathcal{D}_{\mathcal{X}}^{\text{test}} = \mathcal{D}_{\text{query}}$. We consider the setting where the prompt inputs $\mathbf{x}_i$s' scale between training and testing is different. We scale them by a factor of 0.8 or 1.2.

- $\mathcal{D}_{\mathcal{X}}^{\text{train}} = \mathcal{D}_{\mathcal{X}}^{\text{test}} \ne \mathcal{D}_{\text{query}}$. We sample the context examples from the same distribution as at training time, but sample $\mathbf{x}_q$ from a Gaussian distribution with $3\times$ higher standard deviation.

- $\mathcal{D}_{\mathcal{X}}^{\text{test}} \ne \mathcal{D}_{\mathcal{X}}^{\text{train}} = \mathcal{D}_{\text{query}}$. We fix the sign of each coordinate to be randomly positive or negative for all prompt inputs $\mathbf{x}_i$s, and draw $\mathbf{x}_q$ from $\mathcal{N}(0, I)$ as before.

- $\mathcal{D}_{\mathcal{F}}^{\text{train}} \ne \mathcal{D}_{\mathcal{F}}^{\text{test}}$. We consider scaling the weight vector by a factor of 0.8 or 1.2, to capture shifts of task functions.

Through the above diverse evaluation settings, we comprehensively demonstrate that CoQE consistently exhibits stronger ICL capability than a standard Transformer of comparable size on regression tasks.

**Additional results on representation learning.** Our Theorem 5.6 shows that under the dual-space modeling framework, a sufficient set of tasks guarantees a basis-covering sample representation space that the model learns. For empirical validation, we design the task type of two-layer ReLU networks and combination functions, whose different task functions share a common sample representation space in their construction. Figure 5 (a) shows that CoQE indeed achieves a smaller ICL error on these tasks. Furthermore, under the combination functions task, we set the sample representation space dimension of CoQE to 5, matching that of $\Phi$, and directly visualize the 5-dimensional sample representation space learned by CoQE after training (Figure 9). From the figure, we can observe that the five dimensions appear to differentiate in a manner close to the respective transformations of the five dimensions of $\Phi$. Although, due to the equivalence of sample representation spaces under linear transformations, i.e., $f = \mathbf{w}^\top \Phi(\mathbf{x}) = \mathbf{w}^\top H^{-1} \cdot H \Phi(\mathbf{x})$ where $H$ denotes an arbitrary invertible matrix, it is essentially impossible for the model to learn $\Phi$ with perfectly identical scale and shape. The current differentiation can be regarded as another empirical proof of Theorem 5.6 that our dual-modeling could facilitate learning of the basis-covering sample representation space.

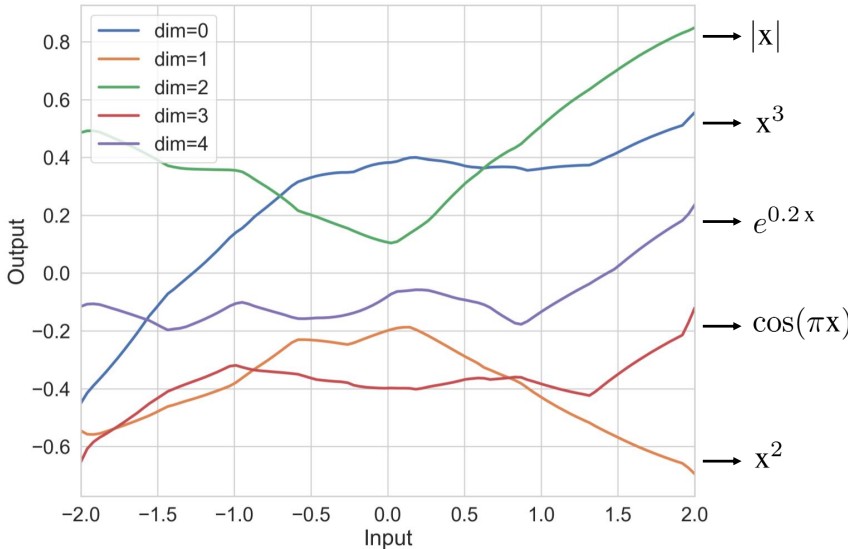

Figure 9: The 5-dimensional sample representation space learned by CoQE for the combination functions.

### B.3 Synthetic Few-shot Classification

**Implementation details.** In our experiments, we employ ResNets of three sizes (with embedding dimensions $E = 64$, $E = 256$ and $E = 512$) to encode images. All architectures consist of four groups, each containing two residual blocks. The difference lies in the embedding dimensions of each group: for the $E = 64$ ResNet, the four groups produce embeddings of sizes 16, 32, 32, and 64, respectively; for the $E = 256$ ResNet, the sizes are 64, 128, 128, and 256; for the $E = 512$ ResNet, the sizes are 64, 128, 256, and 512. Then a fully connected layer is appended to the $E = 256$ and $E = 512$ ResNet to project the final embedding dimension back to 64 before feeding it into the Transformer. Larger embedding dimension of Resnets clearly possesses a stronger capacity for extracting visual sample representations. As a result, the resulting embedding tokens are more expressive. For CoQE, we find that the $E = 64$ ResNet is insufficient for the sample encoder. We also employ two Transformer configurations with different layers: $L = 4$ and $L = 12$. Both variants use an embedding dimension of 64 and 8 attention heads. We have shown that CoQE with only an $L = 4$ Transformer in the task encoder, can match the ICL and IWL performance of an $L = 12$ Transformer. In our experiments, a baseline Transformer with $E = 64$ and $L = 12$ contains approximately 0.9M parameters, CoQE with $E = 256$ and $L = 4$ has 1.0M parameters, while CoQE with $E = 512$ and $L = 4$ has 2.0M parameters.

When training CoQE, we add Gaussian noise to the modified logits to prevent the task encoder's output from collapsing to a static vector. Specifically, the initial noise is sampled from $\mathcal{N}(\mu_0, 1)$, and both the mean and standard deviation are incremented by 1 every $10^4$ training steps. During training of baseline Transformers and CoQE, we use a batch size of 24, a learning rate of $1e-4$, and train for $1 \times 10^5$ steps. All experiments are conducted on $8\times$ NVIDIA V100 GPUs.

**Detailed results of all settings and baselines.** In Table 3, we report the ICL and IWL performance across all model scales, training settings, and algorithms. In Table 4, we report the effect of appropriately scaling both the Transformer and CoQE. These results demonstrates that CoQE can robustly reconcile ICL and IWL

**Ablation study.** We present the training curves of CoQE under different levels of noise (Figure 10). It is evident that, in the absence of noise, the model's ICL capability rapidly decays after an initial emergence, accompanied by a similarly rapid increase in IWL performance. Although this observation is not made under a standard Transformer model, we hypothesize that the underlying phenomenon extends beyond

Table 3: Performance comparison of different Transformer variants and baselines. All values are presented as percentages. The maximum and second-largest values are indicated by boldface and underline, respectively.

| Model | Config | Metric | $(0.9,1)$ | $(0.9,0)$ | $(0.9,2)$ | $(0.5,0)$ | $(0.1,0)$ | Avg. |
|---|---|---|---|---|---|---|---|---|
| | | | \multicolumn{5}{c}{Settings $(P_{\text{bursty}}, \alpha)$} | |
| Transformer | $E=64, L=12$ | ICL | 84.2 | 73.1 | 49.7 | 64.5 | 50.5 | 64.4 |
| | #Param: 0.9M | IWL | 65.1 | 4.1 | 98.0 | 17.8 | 51.5 | 47.3 |
| | $E=64, L=8$ | ICL | 74.4 | 67.2 | 50.2 | 57.5 | 50.3 | 59.9 |
| | #Param: 0.6M | IWL | 65.2 | 8.1 | 97.9 | 21.4 | 57.9 | 50.1 |
| | $E=64, L=4$ | ICL | 50.0 | 50.5 | 50.1 | 50.2 | 50.4 | 50.2 |
| | #Param: 0.4M | IWL | 66.2 | 15.1 | 98.1 | 27.3 | 63.1 | 54.0 |
| | $E=256, L=12$ | ICL | 49.8 | 51.4 | 50.0 | 50.2 | 50.5 | 50.4 |
| | #Param: 1.4M | IWL | 72.3 | 20.1 | 98.4 | 58.1 | 75.4 | 64.9 |
| | $E=256, L=8$ | ICL | 50.8 | 50.7 | 50.2 | 49.9 | 50.7 | 50.5 |
| | #Param: 1.2M | IWL | 73.1 | 28.7 | 98.4 | 60.2 | 78.5 | 67.8 |
| | $E=256, L=4$ | ICL | 50.0 | 50.3 | 50.6 | 49.9 | 50.7 | 50.3 |
| | #Param: 1.0M | IWL | 74.4 | 38.8 | 98.6 | 63.4 | 81.1 | 71.3 |
| | $E=512, L=12$ | ICL | 50.5 | 50.5 | 50.3 | 50.1 | 50.5 | 50.4 |
| | #Param: 2.4M | IWL | 79.6 | 36.0 | 98.9 | 98.4 | 99.0 | 82.4 |
| | $E=512, L=8$ | ICL | 50.4 | 49.8 | 50.3 | 50.2 | 51.0 | 50.3 |
| | #Param: 2.2M | IWL | 81.0 | 49.3 | 99.0 | 98.9 | 99.1 | 85.5 |
| | $E=512, L=4$ | ICL | 49.9 | 50.0 | 51.0 | 49.5 | 51.0 | 50.3 |
| | #Param: 2.0M | IWL | 82.5 | 62.5 | **99.1** | **99.4** | **99.2** | **88.5** |
| Transformer + Noise | $E=64, L=12$ | ICL | 80.0 | 70.3 | 49.9 | 59.8 | 50.1 | 62.0 |
| | #Param: 0.9M | IWL | 60.4 | 1.2 | 91.3 | 8.9 | 49.9 | 42.3 |
| Transformer + Regularization | $E=64, L=12$ | ICL | 80.5 | 64.2 | 50.2 | 67.8 | 53.1 | 63.2 |
| | #Param: 0.9M | IWL | 64.9 | 3.1 | 63.2 | 36.4 | 98.5 | 53.2 |
| Transformer + Temporary Forgetting | $E=64, L=12$ | ICL | 50.4 | 51.0 | 49.8 | 50.5 | 51.0 | 50.5 |
| | #Param: 0.9M | IWL | 54.8 | 0.8 | 97.1 | 7.2 | 38.5 | 39.7 |
| Linear Transformer | $E=64, L=12$ | ICL | 51.3 | 51.0 | 50.0 | 50.1 | 50.3 | 50.5 |
| | #Param: 0.8M | IWL | 68.3 | 8.0 | 98.4 | 21.0 | 53.5 | 49.8 |
| CoQE | $E=256, L=4$ | ICL | 90.3 | 89.2 | 85.8 | 88.0 | 90.3 | 88.7 |
| | #Param: 1.0M | IWL | 82.0 | 81.0 | 63.1 | 82.0 | 87.9 | 79.2 |
| | $E=512, L=4$ | ICL | **92.1** | **91.2** | **88.0** | **90.2** | **92.0** | **90.7** |
| | #Param: 2.0M | IWL | **90.0** | **89.4** | 67.9 | 91.0 | 95.4 | 86.7 |

Table 4: The effect of appropriately scaling both the Transformer and CoQE.

| Model | Config | ICL | IWL | | Model | Config | ICL | IWL |
|---|---|---|---|---|---|---|---|---|
| | $L = 12$ | 84.2 | 65.1 | | | $H = 64$ | 84.2 | 65.1 |
| Trans. $(E = 64, H = 64)$ | $L = 14$ | 86.5 | 62.9 | | Trans. $(E = 64, L = 12)$ | $H = 72$ | 51.0 | 67.4 |
| | $L = 16$ | 73.0 | 62.9 | | | $H = 80$ | 51.0 | 68.1 |
| | $L = 4$ | **90.3** | 82.0 | | | $H = 64$ | 90.3 | 82.0 |
| CoQE $(E = 256, H = 64)$ | $L = 6$ | 89.3 | 84.7 | | CoQE $(E = 256, L = 4)$ | $H = 72$ | **92.6** | **83.0** |
| | $L = 8$ | 89.7 | **85.2** | | | $H = 80$ | 90.1 | 82.2 |

model architecture, reflecting the intrinsic properties of the two strategies. ICL is a lightweight, dynamic strategy, whereas IWL is more training-intensive but ultimately more stable. In standard Transformers, where the two strategies are difficult to co-exist, training often leads to a transition from ICL to IWL. In contrast, CoQE enables robust coexistence of both strategies through explicit modeling and learning of the task representation space, as well as the use of Gaussian noise to isolate the task-transformations associated with each strategy.

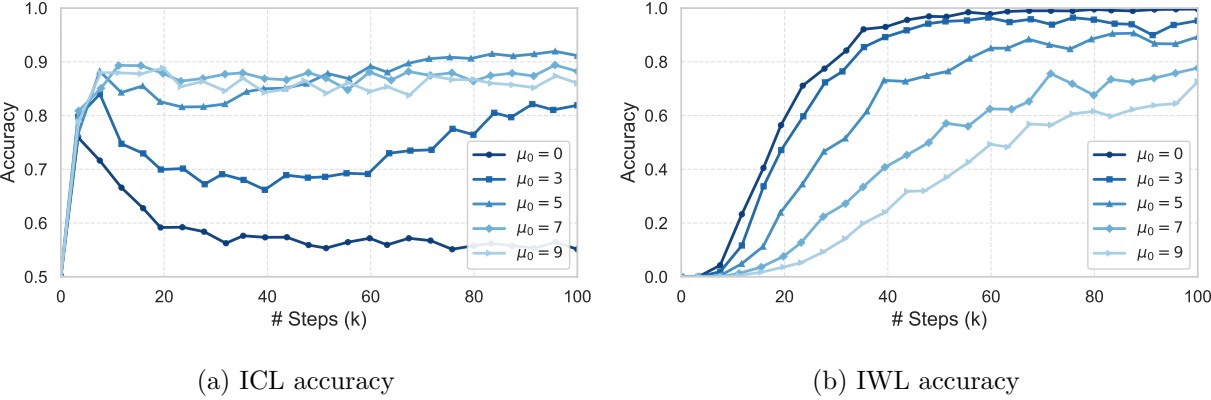

(a) ICL accuracy  (b) IWL accuracy

Figure 10: Learning curves under different noise levels

## B.4 Llama Based Synthetic Few-shot Classification

**Datasets construction.** We saved the token embedding matrices of the Llama-3.2-1B, Llama-3.2-3B, and Llama-3.1-8B models. Each embedding matrix has a shape of $128{,}256 \times d_{\text{model}}$, where $d_{\text{model}} = 2048, 3072$, and 4096 for the 1B, 3B, and 8B models, respectively. To construct "classes" for our few-shot classification experiments from these raw token embeddings, we apply the following three processing steps:

1. Subselection: Among the 128,256 tokens, many correspond to special symbols or non-English text. We selected all tokens consisting solely of English letters A-Z, a-z and underscores, resulting in 33,588 tokens.

2. Clustering: We applied spherical clustering using FAISS, performing 2,400-way clustering and retaining all clusters containing more than 10 tokens. From these, we randomly selected 1,200 clusters.

3. Cluster Sampling: Many clusters contained more than 10 tokens. To maximize intra-class variability, we selected the 10 tokens farthest from each cluster center. Consequently, we obtained 1,200 classes, each containing 10 samples.

Compared with Omniglot images, this construction produces classes with greater intra-class variability while reflecting semantic relations more relevant to NLP tasks. Some sample clusters from Llama-3.2-3B are shown below:

English | edish | apanese | French | Chinese | orean | California | Russian | frican | ustralian

Class | _class | addClass | _CLASS | className | removeClass | classList | Classifier | hasClass | getClass

ellow | green | Green | iolet | orange | urple | agenta | purple | Pink | greens | _YELLOW

**Implementation details.** We trained both the Transformer and CoQE models on the constructed dataset following the same procedure described in the main text, fixing the data distribution parameters to $P_{\text{bursty}} = 0.9$ and Zipfian exponent $\alpha = 1$ to approximate the natural language distribution. In terms of model architecture, for both Transformer and CoQE, we used a three-layer MLP instead of a ResNet as the sample encoder, with hidden dimensions of 512, 256, and 64, respectively. The Transformer consists of 12 layers, while the task encoder in CoQE contains 4 layers, resulting in fewer parameters for CoQE compared with the Transformer. All other training hyperparameters such as learning rate and batch size, are set identical to those used in the image few-shot classification experiments.

