# OpenReview forum: "Reconciling In-Context and In-Weight Learning via Dual Representation Space Encoding"
_TMLR — Accepted by TMLR_

### Review · Reviewer_Q6xq · 2025-12-19

**Summary Of Contributions:**

**Summary**

This paper investigates the trade-off between In-Context Learning (ICL) and In-Weights Learning (IWL) within the Transformer architecture. The authors demonstrate that in a multi-task setting, by introducing a dual-space (referred to as task space), outputs can be decomposed into a scalar product between input representations and a task vector. They argue that standard Self-Attention (SA) modules entangle the representations of the context and the query, which complicates the separation of ICL and IWL mechanisms. To address this, the paper introduces CoQE, a novel architecture that independently encodes the context and the query. In this framework, the output is predicted via a scalar product between the query and a task vector derived from the context. Experimental results suggest that CoQE achieves strong performance for both ICL and IWL tasks.

**Strengths:**

- The "dual-space modeling" offers a theoretically sound and interesting explanation for the entangled structure of representations using standard Self-Attention.
- The proposed CoQE architecture is well-motivated by the theoretical discussion.
- Empirical results, while based on small models, demonstrate good performance across various settings for both ICL and IWL, which is the intended goal.

**Weaknesses:**

- The analysis in Section 4.2 relies only on qualitative visualizations (UMAP) and limited hyperparameter configurations.
- The architectural assumption that queries are separable from context may not hold for all domains (e.g., in text).

**Audience:**

Yes

**Audience Explanation:**

The relationship and trade-off between In-Context Learning (ICL) and In-Weights Learning (IWL) is a central topic in current understanding of the behavior of Transformers. The "dual-space modeling" theoretical framework and the finding that separating context from query processing can improve this balance are valuable insights for researchers working on Transformer interpretability and efficient architecture design.

**Broader Impact Concerns:**

No broader impact concerns.

**Claims And Evidence:**

No

**Claims Explanation:**

While the theoretical derivations are sounded and seem correct, the empirical evidence supporting the analysis in Section 4.2 is currently weak.

1. The first observation relies heavily on qualitative analysis from UMAP plots (Figure 1, bottom). UMAP projections can be sensitive to hyperparameters and misleading regarding global structure. Relying solely on these plots without quantitative metrics (e.g., clustering separation scores) makes the evidence less convincing.
2. The second observation regarding model size is derived from a limited set of experimental settings. Since there is a large combinatorial space of parameters that was not explored during the model size variation experiments (e.g., the $\alpha$ parameter of the Zipfian distribution or $P_{bursty}$), the claims about scaling behavior may be over-extrapolated.

**Requested Changes:**

### Major changes

**Strengthen Section 4.2 Analysis:**
 - Please provide quantitative evidence to support the claims derived from the UMAP plots (e.g., clustering metrics, cosine similarity analysis or measure of distance) rather than relying solely on visuals.
 - Expand the experimental setups regarding model size to include variations in other parameters, and more granularity in current variations, to confirm the findings.

**Discuss the Separability Assumption:** The CoQE architecture relies on a "token-wise" sample encoder that separates the query from the context. While this works for the tasks presented, it is a strong assumption that does not always hold in general, e.g. in NLP where the query is often an inseparable continuation of the context. Please add a paragraph discussing this limitation and how CoQE might (or might not) apply to standard autoregressive language modeling tasks.

### Minor changes

The positioning of Figure 1 breaks the flow of the paper. It is first mentioned on page 1, but referenced multiple times later, requiring the reader to flip back and forth. Please consider moving it closer to the main analysis or splitting it to improve readability.

---

> ### Author Response · Authors · 2026-01-14
>
> We appreciate your comments and suggestions, which have significantly improved our arguments and presentation. Below we provide a point-by-point response.
>
> ### Major 1
>
> To support Observation 1, we employ two quantitative metrics derived from the standard silhouette coefficient, which evaluates cluster cohesion and separation. The silhouette score $s(x)$ measures how similar a sample $x$ is to its own cluster compared to other clusters. Using Euclidean distance $d(\cdot,\cdot)$, we define $a(x)$ as the average distance between $x$ and all other points in its own cluster $C_x$, and $b(x)$ as the average distance to the *nearest* neighboring cluster $C_k$ (where $k \neq x$).
>
> $$
> a(x) = \frac{1}{|C_x| - 1} \sum_{x' \in C_x, x' \neq x} d(x, x'), \quad b(x) = \min_{C_k \neq C_x} \left( \frac{1}{|C_k|} \sum_{x' \in C_k} d(x, x') \right)
> $$
>
> The silhouette score is then given by:
>
> $$
> s(x) = \frac{b(x) - a(x)}{\max\{a(x),\, b(x)\}}
> $$
>
> The first metric we employ is **Context Silhouette Coefficient (CSC)**. It measure whether the model learns "good representations for context." For a fixed object class $c \in \{1,\dots, N\}$ (with $N = 10$), we treat the samples as two distinct clusters based on their **contextual labels** (0 or 1). Here, $a(z)$ measures the distance to samples with the *same* context label, while $b(z)$ measures the distance to the *opposite* context label. We average the scores within each class $c$ and then across all $N$ classes:
> $$
> \text{CSC} = \frac{1}{N} \sum_{c=1}^N\frac{1}{|Z_c|} \sum_{z \in Z_c} s(z).
> $$
>
> The second metric is **Sample Silhouette Coefficient (SSC)**. SSC adapts the metric to measure "good representations for samples" (class separability) without context support. In this setting, the clusters are defined by the $N$ ground-truth object classes. For a sample $i$ with class $y_i$, $a(i)$ is the intra-class distance, and $b(i)$ is the distance to the closest distinct class. The final metric is the average across all $M$ samples of all classes:
> $$
> \text{SSC} = \frac{1}{M} \sum_{i=1}^{M} s(i).
> $$
> In **Figure 2 (left)** of the revised manuscript, we report, for three training settings and four training checkpoints (3k, 10k, 50k, 100k steps), how ICL performance correlates with CSC, and  how IWL performance correlates with SSC. Both show a clear positive correlation, providing quantitative evidence for Observation 1: the structure of the learned representation space is tightly linked to the model’s ICL and IWL behavior.
>
> To strengthen Observation 2, we expand the model-size experiments.  **Figure 2 (right)** presents the effect of varying embedding dimension $E$ (64, 256, 12) on ICL and IWL under 15 combinations of $(P_{\text{bursty}}, \alpha, L)$, and the effect of varying Transformer layers $L$ (4, 8, 12) under 15 combinations of $(P_{\text{bursty}}, \alpha, E)$. The numeric results are provided in the **Table 3**. These experiments validate Observation 2 that larger $E$ improves IWL but causes ICL to disappear,  and larger $L$ consistently enhances ICL while slightly harming IWL. This asymmetric behavior reinforces our conclusion that model capacity affects ICL and IWL differently and highlights the structural nature of the trade-off.

---

> > ### Author Response · Authors · 2026-01-14
> >
> > ### Major2
> >
> > We agree that the relatively strong structural assumptions on the task are a main limitation of our work, as well as of most prior work in this area. Concretely, they induce three potential algorithmic limitations: 1. CoQE assumes that each task instance can be decomposed into an explicit context field and query field. This prevents CoQE from **being directly used for large-scale pre-training on generic text**, in contrast to standard autoregressive language modeling. 2. For tasks where such a context–query decomposition is not available or not well defined, our current algorithm does not yet provide an effective way to apply dual-space modeling. 3. Restriction to non-sequential outputs. Our dual-space formulation currently assumes that the answer $y$ is a value or label rather than a sequence. As a result, CoQE cannot yet handle tasks that require generating free-form natural language as the answer.
> >
> > Regarding the first limitation, we have already explored a concrete extension during the rebuttal process. We constructed a natural-language ICL task and, by modifying the causal attention mask, successfully extended CoQE to an autoregressively pre-trained GPT-2 model. In this setting, we again observe that separating context and query encoding improves ICL performance while preserving IWL performance. The details of this experiment are provided in Section 5 of the revised manuscript. Addressing the other two limitations—by further generalizing the theoretical framework and designing corresponding architectures—will be an important direction for future work. At the same time, we would like to emphasize that structured task design is a common assumption in the current literature on ICL–IWL interaction. Within this scope, we believe our theoretical analysis and algorithmic contributions meaningfully advance the understanding and resolution of the tension between ICL and IWL.
> >
> > ### Minor 2
> >
> > We have moved Figure 1 to appear next to the analysis in Sec 4 in the revised version.

---

### Review · Reviewer_5PHU · 2025-12-21

**Summary Of Contributions:**

This paper studies the conflicting dynamics between in-context learning (ICL) and in-weights learning (IWL), where good performance across both is hard-to-achieve. The paper identifies the shared entangled nature of the encoding in standard softmax-based self-attention architectures as a reason behind conflict. To this end, the paper proposes a novel architecture, called CoQE (Context Query Encoding) that achieves strong empirical performance across the synthetic tasks it considers. The main idea here is to separate IWL from ICL by separating context representations from the query, where the context representations effectively acts as a "context-dependent function" to process the query inputs. The key strengths of this paper is the principled understanding of the underlying ICL-IWL conflict, and the proposal of a novel architecture that provides demonstrable improvements in ICL / IWL capabilities. A key weakness of this paper is the focus on synthetic tasks as opposed to real-world tasks like language modelling, and on small transformer models as opposed large-scale ones used in practice.

**Audience:**

Yes

**Audience Explanation:**

I believe the findings of this paper are likely to be of interest to researchers working on theory of ICL / transformers. The key idea behind this paper are the task-sample decomposition based analysis and the proposal of the CoQE architecture that explicitly maintains a dual space decomposition. Both ideas are interesting and have the potential to influence both (1) future theoretical work in this area, as well as (2) architectural improvements.

**Claims And Evidence:**

Yes

**Claims Explanation:**

The following are the main claims made by the paper:
1. It is difficult to simultaneously achieve good performance in ICL and IWL: This is clearly demonstrated by experiments in Figure 1. I consider the UMAP visualizations to be anecdotal, but the accuracy curves in Figure 1 are more convincing regarding this fact.

2. The conflict between ICL and IWL originates from the difficulty of simultaneously encoding sample information and contextual information within the representation space of Transformers: The paper provides two lines of evidence for this argument. First, there are the UMAP visualizations in Figure 2 which indicate a kind of bimodality in the nature of representations depending on the task parameters. Second, is Theorem 5.9, which demonstrates that under standard SoftMax self attention, there cannot be a task-representation decomposition as is present in Linear Self Attention (Proposition 5.8). Together, these lines of argument are convincing explanations regarding the underlying phenomena in the transformer models considered. It is important also to state that it is unclear whether this still holds for large-scale transformer models, and whether empirically they do learn some dual space representations implicitly, but no claim was made about this in the paper either way.

3. CoQE enhances ICL performance, and successfully reconciles ICL and IWL capabilities: This is also demonstrated convincingly in the experiments in Section 7. The results in Figure 5 clearly show that CoQE is able to outperform the transformer-based model on a wide range of ICL tasks, and also does better out of distribution. Figure 6 and Table 2 also demonstrate its ability to simultaneously achieve strong performance across both ICL and IWL tasks.

One aspect where the paper can improve is that it misses the important baseline of Linear self attention (LSA) while demonstrating its results in Section 7. While proposition 5.8 clearly demonstrates that LSA does have the dual space structure, it is unclear whether SoftMax self-attention is strictly inferior to LSA on the synthetic tasks considered, and how it compares to the proposed CoQE architecture. Regardless of this, the CoQE architecture is still interesting in that it is fundamentally a non-linear architecture as opposed to LSA, which is bilinear.

**Requested Changes:**

One aspect where the paper can improve is to strengthen its experiments, which can help strengthen the claims made, especially that standard transformer architecture have fundamental disadvantages with the ICL-IWL conflict that can be only resolved by novel architectures:

1. As I mention in the claims section of the review, the main change I recommend in the paper is the inclusion of the linear self attention (LSA) baseline to experiments in Section 7.
2. It would also be interesting to see the performance of both transformer models as well as CoQE upon scaling these models, e.g.: by increasing width and depth. It would be interesting to know whether for this task, standard transformer models have a fundamental performance ceiling, or whether at large enough scales, the performance gap might vanish.

With these additional experiments, I believe the underlying phenomena will be better understood (especially as it relates to model scale), and the claims made by the paper can be better extrapolated to large-scale settings.

---

> ### Author Response · Authors · 2026-01-14
>
> We appreciate your comments and suggestions, which have significantly improved the clarity and quality of our manuscript. Below we provide a point-by-point response.
>
> ### Suggestion 1
>
> We have added a linear Transformer as an additional baseline.  In the table below, (0.9, 1) denotes $P_\text{bursty}=0.9, \alpha=1$; others follow similarly.
>
> | Model             |   Config & Params   | Metric | (0.9, 1) | (0.9, 0) | (0.9, 2) | (0.5, 0) | (0.1, 0) |   Avg.   |
> | :---------------- | :-----------------: | :----: | :------: | :------: | :------: | :------: | :------: | :------: |
> | **Transformer**   | $E=64, L=12$ (0.9M) |  ICL   |   84.2   |   73.1   |   49.7   |   64.5   |   50.5   |   64.4   |
> |                   |                     |  IWL   |   65.1   |   4.1    |   98.0   |   17.8   |   51.5   |   47.3   |
> | **Linear Trans.** | $E=64, L=12$ (0.8M) |  ICL   |   51.3   |   51.0   |   50.0   |   50.1   |   50.3   |   50.5   |
> |                   |                     |  IWL   |   68.3   |   8.0    | **98.4** |   21.0   |   53.5   |   49.8   |
> | **CoQE**          | $E=256, L=4$ (1.0M) |  ICL   |   90.3   |   89.2   |   85.8   |   88.0   |   90.3   |   88.7   |
> |                   |                     |  IWL   |   82.0   |   81.0   |   63.1   |   82.0   |   87.9   |   79.2   |
> |                   | $E=512, L=4$ (2.0M) |  ICL   | **92.1** | **91.2** | **88.0** | **90.2** | **92.0** | **90.7** |
> |                   |                     |  IWL   | **90.0** | **89.4** |   67.9   | **91.0** | **95.4** | **86.7** |
>
> We observe that its ICL performance is significantly worse than that of the standard Transformer, which we attribute to the limited expressive power of the LSA structure. In Proposition 5.8, our discussion of how LSA supports dual-space modeling is based on a simplified setting where all task nonlinearity is captured by the feature encoder, following ICL modeling framework in [1]. However, in real tasks, both the sample representation and the task representation are produced by complex nonlinear processes. **Therefore, as you pointed out, the main contribution of the CoQE architecture lies in fully exploiting the expressive capacity of nonlinear networks to realize a dual-space structure, thereby effectively reconciling ICL and IWL capabilities in practice.**
>
> ### Suggestion 2
>
> We also evaluated the effect of appropriately scaling both the Transformer and CoQE. The impact of increasing the width of the ResNet component has already been discussed in Observation 2 in Sec 4.2. Here, we further report results for scaling up the Transformer part by increasing the number of layers ($L$) and the hidden state dimension ($H$):
>
> | Model                          | Config |   ICL    |   IWL    |
> | :----------------------------- | :----: | :------: | :------: |
> | **Transformer** ($E=64, H=64$) | $L=12$ |   84.2   |   65.1   |
> |                                | $L=14$ |   86.5   |   62.9   |
> |                                | $L=16$ |   73.0   |   62.9   |
> | **CoQE** ($E=256, H=64$)       | $L=4$  | **90.3** |   82.0   |
> |                                | $L=6$  |   89.3   |   84.7   |
> |                                | $L=8$  |   89.7   | **85.2** |
>
>
>
> | Model                          | Config |   ICL    |   IWL    |
> | :----------------------------- | :----: | :------: | :------: |
> | **Transformer** ($E=64, L=12$) | $H=64$ |   84.2   |   65.1   |
> |                                | $H=72$ |   51.0   |   67.4   |
> |                                | $H=80$ |   51.0   |   68.1   |
> | **CoQE** ($E=256, L=4$)        | $H=64$ |   90.3   |   82.0   |
> |                                | $H=72$ | **92.6** | **83.0** |
> |                                | $H=80$ |   90.1   |   82.2   |
>
> The results suggest that "standard Transformer models have a fundamental performance ceiling". Moreover, when the hidden state dimension becomes larger, we observe a phenomenon similar to that in Observation 2 with increased ResNet width: the model loses its ICL capability. Our interpretation is again that when a single token becomes sufficiently expressive to solve the task, the model tends to ignore the context. In contrast, CoQE consistently maintains superior ICL–IWL trade-offs under scaling. This demonstrates the potential of our dual-space–motivated architectural prior to transfer effectively to larger-scale settings.
>
> [1] Kim et al. Transformers Learn Nonlinear Features In Context: Nonconvex Mean-field Dynamics on the Attention Landscape. ICML2024

---

### Review · Reviewer_3mWf · 2026-01-03

**Summary Of Contributions:**

This paper examines the ICL–IWL trade-off in Transformer-like models. It introduces a dual-space view where a task is represented as a linear functional over sample representations (via the Riesz theorem), argues that softmax self-attention entangles context and query encoding, and proposes CoQE to separate them by encoding context into a task vector and queries into sample vectors combined via an inner product. Experiments on regression and few-shot classification (plus a token-embedding classification proxy) show improved ICL/IWL trade-offs.

Strengths:

S1. The dual-space viewpoint provides a clean conceptual lens connecting task inference from context to linear readout over sample representations.

S2. The proposed architecture is simple and interpretable, with a clear mapping from the motivating framework to an implementable design.

S3. Results suggest an improved ICL/IWL trade-off on the included benchmarks under the paper’s evaluation protocol.

Weaknesses:

W1. Evidence for extension to open-ended autoregressive generation is limited; the NLP-facing experiment remains a classification proxy rather than a generative setting.

W2. In few-shot classification, gains appear closely tied to noise injection, making it hard to separate “architecture benefit” from “regularization knob” without stronger controls.

W3. The benchmark suite and baseline set remain relatively limited, leaving uncertainty about robustness on more complex tasks and against stronger alternatives.

W4. The linear representation hypothesis is a meaningful but restrictive assumption; beyond being acknowledged, it limits the scope of the conclusions without additional validation.

**Audience:**

Yes

**Audience Explanation:**

The ICL–IWL interaction is an active topic in ML community.

**Claims And Evidence:**

No

**Claims Explanation:**

The paper provides clear evidence that CoQE improves the reported ICL/IWL metrics on the included regression and few-shot classification settings. However, broader implications e.g. extension to open-ended generation and attribution of improvements to the architectural separation rather than noise/regularization are not fully supported by the current experiments and controls.

**Requested Changes:**

Suggestions:

C1. Add an experiment in an autoregressive setting (even a small-scale next-token or conditional generation task) to substantiate claims about broader NLP applicability, or otherwise clearly explain the intended scope and why generation is out of scope.

C2. Strengthen the causal story by adding controls to help disentangle architecture vs. noise/regularization effects (e.g., apply comparable noise/regularization to the Transformer baseline, or provide additional ablations), or explain why such controls are not feasible.

C3. Broaden evaluation modestly (one more complex or longer-context task, or a more challenging synthetic/compositional benchmark) and/or add one or two stronger baselines; if not possible, clarify what conclusions should and should not be drawn from the current task set.

---

> ### Author Response · Authors · 2026-01-14
>
> We are grateful for your valuable suggestions and constructive criticism. Below we provide a point-by-point response.
>
> ### Suggestion 1
>
> In analogy to the autoregressive experiment in [1], we designed a small-scale generation task to further evaluate our method. We refer it as **Conditional Pseudo-arithmetic Task**.
>
> We construct sequences consisting of arithmetic conditions followed by a word problem. For example:
>
> > *“Condition: 15 + 23 = 38. 43 + 43 = 86. 17 + 5 = 22. Question: A farmer has 43 apples and buys another 43 ones, now how many apples does he have in total? Answer: 86.”*
>
> Here, the question can be answered solely based on the given conditions. We fine-tune GPT-2 on 5,000 such sequences and evaluate its accuracy on the same distribution, which we interpret as its **IWL performance**.
>
> After fine-tuning, we then construct sequences that mix standard arithmetic with “pseudo-arithmetic” over letters, for example:
>
> > *“Condition: 5 + 23 = 28. Y + F = C. 30 + 35 = 65. Question: The store sold Y pencils in the morning and F in the afternoon. Total pencils sold? Answer: C.”*
>
> In this case, the model must rely on the contextual relation given in the sequence (rather than memorized weights) to answer correctly, which we use to measure **ICL performance**. In this setup, $x_i$ and $x_q$ are generalized from single tokens (as in the main experiments) to sequence-level inputs.
>
> We extend CoQE to the pretrained GPT-2 as follows. During fine-tuning, we control the attention mask so that tokens in the *Question* field cannot attend to the Condition field, thereby implementing **separate Context–Query Encoding**. On the output side, we largely follow the “Implementation for Few-shot Classification” in the main paper. The difference is that we do not introduce $L_{\text{orig}}$ or noise injection, because we observe that fine-tuning pretrained GPT-2 on this dataset is fast and stable. Besides standard fine-tuning, we introduce two more baselines that were proposed to alleviate the tension between ICL and IWL: L2 regularization [2] and Probabilistic Temporary Forgetting [1]. All models are fine-tuned for 300 steps. The results are summarized in the table below and in **Figure 7** of the revised manuscript.
>
> | Method                   |  ICL  |  IWL  |
> | :----------------------- | :---: | :---: |
> | **Vanilla Fine-Tuning**  | 71.3  | 100.0 |
> | **Regularization**       | 65.3  | 100.0 |
> | **Temporary Forgetting** | 20.8  | 100.0 |
> | **CoQE**                 | 100.0 | 99.8  |
>
> After 300 steps of fine-tuning, all methods achieve nearly 100% IWL performance on the training-distribution evaluation, while CoQE achieves substantially higher ICL accuracy than the other three baselines on the pseudo-arithmetic evaluation. A typical failure mode we observe is using the wrong condition, for example:
>
> > *“Condition: 42 + 4 = 46. 31 + 20 = 51. L + Y = E. Question: I read L pages yesterday and Y pages today. How many pages did I read? Answer: 51.”*
>
> This indicates that the model is still limited in leveraging the correct in-context relation in this scenario. Moreover, Regularization and Probabilistic Temporary Forgetting perform even worse than vanilla fine-tuning. This demonstrate limitations of their design: Regularization can take effect at very large training steps (e.g., $10^7$), and Probabilistic Temporary Forgetting targets ICL settings that involve random-vector inputs, which differs from our structured pseudo-arithmetic setting. Overall, this simple autoregressive experiment further validates the effectiveness and transferability of CoQE beyond the original classification scenario.
>
> On the other side, we acknowledge that this is **not a  open-ended generation task**: although $x$ is extended to a sequence, the answer $y$ remains a single token rather than a natural language span. This reflects a key limitation of our current dual-space modeling framework: we implicitly assume that $y$ is a value or label, not a sequence. Extending the theoretical modeling and architectural design to handle general open-ended NLP generation tasks is an important direction for future work. At the same time, we note that structured task design is a common assumption in the literature on ICL–IWL interaction; to the best of our knowledge, existing work typically also focuses on non–open-ended generation. We believe that our theoretical framework and algorithm provide meaningful progress toward understanding and mitigating the tension between ICL and IWL.

---

> > ### Author Response · Authors · 2026-01-14
> >
> > ### Suggestion 2 & Suggestion 3
> >
> > To verify whether the benefits of CoQE arise solely from noise injection, we applied comparable noise directly to the logits of a standard Transformer, using a procedure similar to that in CoQE. The results in the table below show that simply injecting noise does **not** alleviate the tension between ICL and IWL in Transformers. In our framework, noise injection is used only as a practical trick within the dual-space modeling algorithm, designed to prevent the task representation space constructed by the task encoder from collapsing during training. Its effectiveness illustrates the value of maintaining a dynamic and rich task representation space, which supports the theoretical motivation of our model. At the same time, the additional experiments in Suggestion 1 demonstrate that noise injection is not essential: the key algorithmic insight we obtain is the separation of context and query encoding.
> >
> > Beyond the conditional autoregressive setting added in Suggestion 1, we also include Regularization and Probabilistic Temporary Forgetting as additional baselines in our original few-shot classification experiments. Both methods were proposed as approaches to mitigate the tension between ICL and IWL. However, our results show that they do not significantly outperform directly training the Transformer, which highlights the limitations of these algorithms. In the table below, (0.9, 1) denotes $P_\text{bursty}=0.9, \alpha=1$; others follow similarly.
> >
> > | Model                   |   Config & Params   | Metric | (0.9, 1) | (0.9, 0) | (0.9, 2) | (0.5, 0) | (0.1, 0) |   Avg.   |
> > | :---------------------- | :-----------------: | :----: | :------: | :------: | :------: | :------: | :------: | :------: |
> > | **Transformer**         | $E=64, L=12$ (0.9M) |  ICL   |   84.2   |   73.1   |   49.7   |   64.5   |   50.5   |   64.4   |
> > |                         |                     |  IWL   |   65.1   |   4.1    | **98.0** |   17.8   |   51.5   |   47.3   |
> > | **Trans. + Noise**      | $E=64, L=12$ (0.9M) |  ICL   |   80.0   |   70.3   |   49.9   |   59.8   |   50.1   |   62.0   |
> > |                         |                     |  IWL   |   60.4   |   1.2    |   91.3   |   8.9    |   49.9   |   42.3   |
> > | **Trans. + Reg.**       | $E=64, L=12$ (0.9M) |  ICL   |   80.5   |   64.2   |   50.2   |   67.8   |   53.1   |   63.2   |
> > |                         |                     |  IWL   |   64.9   |   3.1    |   63.2   |   36.4   | **98.5** |   53.2   |
> > | **Trans. + Forgetting** | $E=64, L=12$ (0.9M) |  ICL   |   50.4   |   51.0   |   49.8   |   50.5   |   51.0   |   50.5   |
> > |                         |                     |  IWL   |   54.8   |   0.8    |   97.1   |   7.2    |   38.5   |   39.5   |
> > | **CoQE**                | $E=256, L=4$ (1.0M) |  ICL   |   90.3   |   89.2   |   85.8   |   88.0   |   90.3   |   88.7   |
> > |                         |                     |  IWL   |   82.0   |   81.0   |   63.1   |   82.0   |   87.9   |   79.2   |
> > | **CoQE**                | $E=512, L=4$ (2.0M) |  ICL   | **92.1** | **91.2** | **88.0** | **90.2** | **92.0** | **90.7** |
> > |                         |                     |  IWL   | **90.0** | **89.4** |   67.9   | **91.0** |   95.4   | **86.7** |
> >
> >
> >
> > [1] Anand et al. Dual Process Learning: Controlling Use of In-Context vs. In-Weights Strategies with Weight Forgetting. ICLR2025
> >
> > [2] Singh et al. The Transient Nature of Emergent In-Context Learning in Transformers. NeurIPS2023

---

### Author Response · Authors · 2026-01-14
**General response for all reviewers**

We sincerely appreciate all the reviewers’ careful evaluation and insightful comments, and we have the manuscript accordingly.

The main revisions are as follows:

- **Sec. 4.2:** We added **quantitative evidence** to further support *Observation 1*, and included additional experimental setups regarding model size to further support *Observation 2*. A new Figure 2 has also been added (Reviewer Q6xq).
- **Sec. 7.4:** We added the **Conditional Pseudo-Arithmetic Task** to further verify the generalization ability of our algorithm on autoregressive generative tasks.  A new Figure 7 has also been added (Reviewer 3mWf).
- **Sec. 8:** We further expanded the discussion on the theoretical and algorithmic **limitations** (Reviewer Q6xq; Reviewer 3mWf).
- **Ablation B.3:** We added **Table 3**, which includes results under different training parameters, model sizes, and four more baselines (noise, regularization,  probabilistic temporary forgetting and linear-transformer) (Reviewer Q6xq; Reviewer 5PHU; Reviewer 3mWf).
- **Ablation B.3:** We added **Table 4**, which includes width and depth scaling results for the Transformer and CoQE (Reviewer 5PHU).

---

### Decision · Action_Editor_8zyD · 2026-02-04

**Recommendation:** Accept with minor revision

**Additional Comments:**

Please revise the abstract slightly to
1) state clearly the limitations of the approach with respect to single-token answers vs. the autoregressive case
2) be more explicit about the range of settings evaluated in the final sentence "successfully reconciles ICL and IWL capabilities across all tested conditions."

**Audience:**

Yes

**Audience Explanation:**

The reviewers and I agree that the topic area — the tradeoff between ICL and IWL — is very timely, and the approach offers an interesting use of representational analyses to propose a model improvement, and thus will be of interest to several subsets of readers, despite the limitations of the experiments.

**Claims And Evidence:**

Yes

**Claims Explanation:**

There were concerns from several reviewers about the narrowness of the initial experiments relative to the claims being made; the authors addressed these both by expanding their experiments and refining the discussion of important limitations, such as not addressing the autoregressive case. Thus, I believe that the claims are refined in the paper. I will request that the authors also briefly state these limitations in the abstract, however, when preparing the final version of the work.

---

> ### Author Response · Authors · 2026-02-21
> **Revision in the Camera-ready Version**
>
> Dear Action Editors,
>
> We sincerely appreciate your efforts and constructive feedback during the review process. Based on your suggestions, we have made the following revisions in the camera-ready version:
>
> - We revised the abstract to clarify the single-value task setting and the precise experimental scope.
> - We updated other statements in the main text related to the experimental scope to make them more specific.
> - We improved the presentation of Figure1.
>
> Best,
> Authors